# Identifiability Challenges in Sparse Linear Ordinary Differential Equations

**Cecilia Casolo, Sören Becker, Niki Kilbertus**
Technical University of Munich
Helmholtz Munich
Munich Center for Machine Learning (MCML)
`first.last@helmholtz-munich.de`

## Abstract

Dynamical systems modeling is a core pillar of scientific inquiry across natural and life sciences. Increasingly, dynamical system models are learned from data, rendering identifiability a paramount concept. For systems that are not identifiable from data, no guarantees can be given about their behavior under new conditions and inputs, or about possible control mechanisms to steer the system. It is known in the community that "linear ordinary differential equations (ODE) are almost surely identifiable from a single trajectory." However, this only holds for dense matrices. The sparse regime remains underexplored, despite its practical relevance with sparsity arising naturally in many biological, social, and physical systems. In this work, we address this gap by characterizing the identifiability of sparse linear ODEs. Contrary to the dense case, we show that sparse systems are unidentifiable with a positive probability in practically relevant sparsity regimes and provide lower bounds for this probability. We further study empirically how this theoretical unidentifiability manifests in state-of-the-art methods to estimate linear ODEs from data. Our results corroborate that sparse systems are also practically unidentifiable. Theoretical limitations are not resolved through inductive biases or optimization dynamics. Our findings call for rethinking what can be expected from data-driven dynamical system modeling and allows for quantitative assessments of how much to trust a learned linear ODE.

## 1 Introduction and related work

The field of dynamical systems has emerged early as a corner stone of scientific modeling, primarily due to the high demand for modeling temporally evolving systems in the natural and life sciences. For the most part, dynamical systems have been modeled "manually," i.e., by humanly-prescribed differential equations or simulators inspired by, and validated in real-world systems. With the advent of machine learning and large data collection efforts, dynamical systems are increasingly learned from data. This new perspective also brought about a shift from an original focus on the *forward problem*—solving given differential equations from different initial conditions for predictions—to the *inverse problem*—finding the governing differential equation from observed trajectories. Inferring the underlying dynamical laws governing a system solely from observational data is a longstanding aspiration across numerous scientific disciplines. A fundamental prerequisite to realize this goal is the identifiability of the governing laws, i.e., that the true dynamics can, in principle, be uniquely determined from the available observations. If multiple different laws could result in the exact same observations, we are stuck. This is the problem of identifiability, as illustrated in Fig. 1: could the observed data have arisen from only a single unique dynamic? While solving the inverse problem poses a host of practical challenges, no guarantees can ever be given for unidentifiable systems.

The identifiability of different types of dynamical systems from different types of observed data has been studied in a variety of domains like natural science (Donà et al., 2022; Muñoz-Tamayo et al., 2018), control theory (Ding & Toulis, 2020; Gargash & Mital, 1980), experimental design (Raue et al., 2010) and many more. Miao et al. (2011) provide a well-structured overview of how identifiability analysis of different types of (non-linear) dynamics is approached in practice while Scholl et al. (2022;

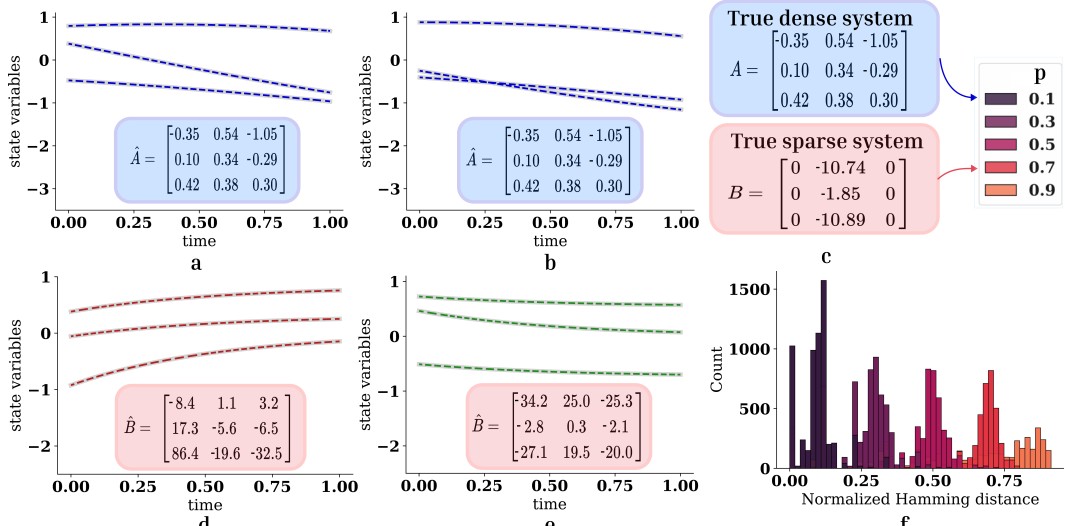

Figure 1: Identifiability differs between dense and sparse systems. The trajectories in **a** and **b** correspond to solutions for two different initial conditions of the dense system in **c**; similarly, trajectories in **d** and **e** correspond to solutions of the sparse system in **c**. All estimated trajectories (colored dashed curves) fit the respective observations (thick gray curves) well, yet only the estimate of the dense system is correct. The histogram in **f** shows that the normalized Hamming distance between the estimated and true system matrix systematically increases as sparsity $p$ increases.

2023) have characterized necessary and sufficient conditions about what needs to be observed for identifiability in different functional classes of dynamics. The bio-mathematics community has been among the first to develop rigorous approaches to study identifiability (Bellman & Åström, 1970), referring to it as "structural identifiability." Similarly, researchers in epidemiology have a strong track record in analyzing identifiability for domain-specific compartment models (Saccomani, 2011; Miao et al., 2011; Xia & Moog, 2003; Tuncer et al., 2016). Cunniffe et al. (2024) recently analyzed the connection between identifiability and observability, as a critical aspect of epidemiological models, where we typically cannot observe the entire state. Stochastic differential equations often enjoy stronger theoretical identifiability guarantees, typically because the stochastic component is assumed to have full support, allowing to "probe the entire space" (Bellot et al., 2021; Wang et al., 2023).

Irrespective of theoretical considerations, new practical methods for learning dynamical systems from data are proposed continuously. These range from traditional parameter estimation techniques for ODEs (Lavielle et al., 2011; Commenges et al., 2011; Huang et al., 2006; Li et al., 2005; Brunton et al., 2016), to neural-network-based parameter estimation (Rubanova et al., 2019; Qin et al., 2019) as well as deep-learning based approaches that learn the dynamics as a neural net (Chen et al., 2018) or predict symbolic expressions (Becker et al., 2023; d'Ascoli et al., 2024). While these methods are reported to achieve strong *reconstruction performance*, i.e., they find dynamics that, when integrated from the same initial condition, recover the observed trajectories well, it is rarely reported to which extent they actually *identify the originally underlying dynamical law*. Due to unidentifiability, these methods could learn dynamical systems that do not generalize beyond the observed time spans or to new initial conditions, let alone warrant claims of scientific insights.

In this work, we will focus on linear, homogeneous, autonomous ordinary differential equations (ODE), $\dot{x}(t) = Ax(t)$, heavily relied upon in many domains while also amenable to rigorous theoretical analysis. Pioneering works by Stanhope et al. (2014) and, more recently, Qiu et al. (2022) established as a key result that "almost all such linear ODEs are identifiable from a single solution trajectory." Here, it is assumed that we have observed the entire trajectory without any observation noise in continuous time. This is often paraphrased as "linear ODEs are identifiable" and summoned by practitioners as justification to assume that whenever a linear ODE is estimated well from data, one has actually found the unique underlying governing law of the system with probability 1. The core contributions in this work build on the observation that this, rightfully celebrated, result only holds for dense matrices, i.e., they assume a measure over systems $\mathbb{R}^{n \times n}$ that is absolutely continuous with respect to the $n$-dimensional Lebesgue measure $\lambda^n$.

This is not satisfied by sparse systems. Sparsity in dynamical systems means that not all variables depend on (or interact with) all other variables (Aliee et al., 2021; 2022). Hence, sparsity allows

us to understand and interpret complex networks pervasive in nature, society, and technology. In fact, dynamical models in biology (Lu et al., 2011) or social networks (Ravazzi et al., 2017) typically exhibit high degrees of sparsity. For example, Liben-Nowell (2005) asserts that "on average, a person has on the order of at most a thousand friends." Similarly, interactions in gene regulatory networks are known to be sparse, with only a tiny fraction of possible pair-wise interactions being nonzero (Carey et al., 2018). Hence, in many settings researchers only attempt to learn dynamical models from data when they assume interactions to be sparse, because meaningful interpretation is only possible in sparse models in the first place (Xu et al., 2023). The resulting models are also believed to have a lower risk of overfitting, particularly when data is limited (Bartoldson et al., 2020).

**Contributions.** This work puts an asterisk on the result that "almost all linear ODEs are identifiable" proving positive lower bounds on the probability of *sparse systems* being unidentifiable. We also quantify theoretically to what extent "near unidentifiability" poses a challenge for practical identification in identifiable cases. Finally, we bridge the gap from theory to practice and demonstrate that (near) unidentifiability is not circumvented in practical methods by potential inductive biases or optimization dynamics. Our work clearly characterizes the regimes (in terms of dimensionality and sparsity) in which unidentifiability is an issue, and what observed time horizons are required to escape near unidentifiability.

## 2 BACKGROUND AND PROBLEM SETTING

We focus on autonomous, homogeneous, linear, noise-free ordinary differential equations (ODE), i.e., initial value problems (IVP) of the form

$$\dot{\boldsymbol{x}}(t) := \frac{d\boldsymbol{x}(t)}{dt} = A\boldsymbol{x}(t), \qquad \boldsymbol{x}(0) = \boldsymbol{x}_0, \tag{1}$$

where $A \in \mathbb{R}^{n \times n}$ is also called the *system* or simply referred to as the "ODE," $\boldsymbol{x} : [0, T] \to \mathbb{R}^n$ denotes the *trajectory*, $\boldsymbol{x}_0 \in \mathbb{R}^n$ is the *initial condition*, and $t$ denotes time. In autonomous systems, we can assume without loss of generality that the initial time is $t = 0$ as autonomy implies that the system $A$ is not time-dependent. Homogeneity ensures that the dynamics are not externally forced or perturbed. The system matrix $A$ is also referred to as the *adjacency matrix* following a graph theoretic view on the dynamics. The key novelty in our work is in considering *sparse* systems, i.e., sparse matrices $A$.

**Discussion of assumptions and limitations.** Linearity, autonomy, homogeneity and the absence of noise heavily restrict the types of systems studied in our work. The linearity assumption is in line with existing work (Stanhope et al., 2014; Qiu et al., 2022; Fedoryuk, 2012; Ovchinnikov et al., 2022) and required for the feasibility of theoretical analysis. Autonomy and homogeneity are adopted to represent our focus on passive observations of naturally evolving systems without external forcing, which corresponds closely to the idea of an observational distribution in the causal inference literature and differentiates our work from approaches towards controllability/observability in the (optimal) control and system identification literature. Some of the existing work on identifying linear ODEs from a single trajectory have been extended to post-nonlinear models (breaking linearity) (Miao et al., 2011), affine systems (extending homogeneity) (Duan et al., 2020), systems with hidden confounder Wang et al. (2024a), and include discrete, noisy observations (Wang et al., 2024b). We believe these works offer fruitful pointers towards extending our analysis of sparse systems here beyond its current limitations. At the same time, Scholl et al. (2023) show that in more general non-linear function classes identifiability requires observing trajectories that essentially "cover the entire state space", putting theoretical limits on what one can hope for in generic non-linear settings. More implicitly, we also assume full observations, i.e., that the entire state relevant for the evolution of the system can be observed. In many practical settings, one may be limited by partial observability. Existing results indicate that identifiability is generally impossible in such settings without strong assumptions on the observation function (Cobelli & Romanin-Jacur, 2007), which is why partial observations are beyond the scope of this work. We discuss our assumptions and their implications in detail in Section A.

**Main goal.** Our main goal is to answer the following question: Given a trajectory $\boldsymbol{x} : [0, T] \to \mathbb{R}^n$ that solves the IVP in Eq. (1), (when) can we uniquely infer $A$ under the assumption that $A$ is sparse?

Let us first formalize and concretize this identifiability problem.

**Definition 1** (identifiability from $\boldsymbol{x}$ in $\Omega \subseteq \mathbb{R}^{n \times n}$). *Let $A \in \Omega \subseteq \mathbb{R}^{n \times n}$ and let $\boldsymbol{x} : [0, T] \to \mathbb{R}^n$ be a solution of $\dot{\boldsymbol{x}}(t) := \frac{d\boldsymbol{x}}{dt}(t) = A\boldsymbol{x}(t)$. We call $A$ identifiable from $\boldsymbol{x}$ in $\Omega$ if there exists no $B \in \Omega$ with $B \neq A$ and $\frac{d\boldsymbol{x}}{dt}(t) = B\boldsymbol{x}(t)$.*

**Remarks.** The IVP in Eq. (1) has a unique solution $\boldsymbol{x}$ on all of $\mathbb{R}$ for all $A$ and $\boldsymbol{x}_0$ by the Picard-Lindelöf theorem and the fact that linear functions are globally Lipschitz (Arnold, 1992). Therefore, instead of $[0, T]$ we can use any interval (including all of $\mathbb{R}$) as time interval in Definition 1. In this work, we are specifically interested in identifying *A for a given observed trajectory*. Related but different notions of identifiability studied in the literature include, for example (Stanhope et al., 2014; Qiu et al., 2022): (a) For a fixed initial condition $\boldsymbol{x}_0 \in \mathbb{R}^n$, all pairs of distinct systems $A, B \in \Omega$ lead to different trajectories. (b) For all pairs of distinct systems $A, B \in \Omega$, there exists some initial condition $\boldsymbol{x}_0 \in \mathbb{R}^n$ leading to different trajectories. (c) For all pairs of distinct systems $A, B \in \Omega$, every initial condition $\boldsymbol{x}_0 \in \mathbb{R}^n$ leads to distinct trajectories. It is easy to see that notion (b) is true "globally," i.e., even for $\Omega = \mathbb{R}^{n \times n}$ (Stanhope et al., 2014). This relates closely to the typical linear time invariant (LTI) setting in system identification including controls: when we can "probe" the system by choosing different initial conditions as controls, we are always able to identify it fully. Instead, we concretely focus on identifiability from merely observational data in the form of a single solution trajectory. On the flip side, (c) is not true for $\Omega = \mathbb{R}^{n \times n}$ (and does not even hold for "most" $\Omega \subseteq \mathbb{R}^{n \times n}$, see Qiu et al., 2022, Sec. S1). This is primarily due to the existence of "unlucky" initial conditions, which we will make rigorous below.

To decide when a system can be identified from a given trajectory, it is useful to characterize whether there are systems for which this is never possible. Throughout this work, we will primarily focus on the setting where $\Omega = \mathbb{R}^{n \times n}$ reflecting the fact that we typically cannot exclude any systems from potentially being the ground truth, while still allowing for different probability measures on $\Omega$.

**Definition 2** (global unidentifiability). *We call $A \in \mathbb{R}^{n \times n}$ globally unidentifiable in $\Omega$, if there exists no $\boldsymbol{x}_0 \in \mathbb{R}$ such that $A$ is identifiable from the corresponding solution trajectory $\boldsymbol{x}$ in $\mathbb{R}^{n \times n}$. We denote the set of all globally unidentifiable $A$ by $\mathcal{U} \subseteq \mathbb{R}^{n \times n}$.*

In words, a system $A$ is globally unidentifiable if any solution trajectory is also the solution to some other system $B$ (this need not be a single alternative $B$, but for each trajectory there must exist at least one). A necessary and sufficient condition for global unidentifiability that characterizes these "hopeless" systems reads as follows.

**Theorem 1** (Stanhope et al., 2014, Thm 2.5 + Thm 3.6). *A system $A \in \mathbb{R}^{n \times n}$ is globally unidentifiable if and only if it has more than one Jordan block corresponding to the same eigenvalue (for at least one eigenvalue) in the Jordan normal form of $A$.*

Additionally, they also characterized when $A$ is identifiable from $\boldsymbol{x}$.

**Theorem 2** (Stanhope et al., 2014, Lem 3.4 + Thm 3.4). *A system $A \in \mathbb{R}^{n \times n}$ is identifiable from $\boldsymbol{x}$ if and only if $\boldsymbol{x}(0)$ is not contained in any $A$-invariant proper subspace of $\mathbb{R}^n$.*

Whenever any point along the solution trajectory (e.g., the initial condition) lies in an $A$-invariant *proper* subspace of $A$, the entire trajectory will be confined to this subspace. The trajectory is thus not "probing" the complement of this subspace in $\mathbb{R}^{n \times n}$. We can then construct another system $B$ that agrees with $A$ (viewed as linear maps) on the $A$-invariant subspace, but differs on the complement.

## 3 SYSTEM-LEVEL UNIDENTIFIABILITY

We will refer to $A$ being in $\mathcal{U}$ as "system level unidentifiability," because when $A \in \mathcal{U}$ it is unidentifiable regardless of the observed trajectory. For all $A \notin \mathcal{U}$, there exists at least one solution trajectory $\boldsymbol{x}$, such that $A$ is identifiable from $\boldsymbol{x}$ in $\mathbb{R}^{n \times n}$. However, there may still be "unlucky" $\boldsymbol{x}$, namely those confined to an $A$-invariant proper subspace, for which $A$ remains unidentifiable. For any system $A$, we define $\mathcal{U}_A \subseteq \mathbb{R}^n$ to be the set of initial conditions $\boldsymbol{x}_0 \in \mathbb{R}^n$ from which $A$ is not identifiable. We call this "trajectory level unidentifiability".

Ultimately, our main goal aims at quantifying whether we should expect to be able to identify "a random system" from "a random solution trajectory." To formalize this question, consider a joint probability distribution $P_{A, \boldsymbol{x}_0}$ over pairs of systems and initial conditions $(A, \boldsymbol{x}_0) \in \mathbb{R}^{n \times n} \times \mathbb{R}^n$

(equipped with the Borel-$\sigma$-algebra). We are then interested in

$$P_{A,\boldsymbol{x}_0}(A \text{ not identifiable from } \boldsymbol{x}_0) = \int P_{A,\boldsymbol{x}_0}(\mathcal{U}_A \mid A)\, dP_A(A) \tag{2}$$

$$= P_A(\mathcal{U}) + \int \chi_{A \notin \mathcal{U}}(A) P_{A,\boldsymbol{x}_0}(\mathcal{U}_A \mid A)\, dP_A(A)\,,$$

where $\chi$ is the indicator function, i.e., $\chi_{A \notin \mathcal{U}}(A) = 1$ if $A \notin \mathcal{U}$ and $0$ otherwise. The partitioning in the last equality corresponds to a two-step reasoning: What is the probability of hitting a globally unidentifiable system (on the system identification level) and, separately, for any non globally unidentifiable model, what is the probability of hitting an "unlucky" initial condition.[1]

Before developing a lower bound for Eq. (2), we introduce the assumed sparsity model.

**Definition 3** (Sparse-Continuous Ensemble). *We call a matrix $A := (B_{ij} X_{ij})_{i,j=1}^n$ with $B_{ij} \sim \text{Ber}(1-p)$ independent Bernoulli random variables for some $p \in [0,1]$, and $X_{ij} \sim P_X$ independent continuous random variables (i.e., their distribution is absolutely continuous with respect to the Lebesgue measure on $\mathbb{R}$), a* sparse-continuous *random matrix (or system) with sparsity level $p$.*

Examples include any iid sub-Gaussian random matrix masked by iid Bernoulli variables.

**Lemma 1.** *For a sparse-continuous random matrix $A$, the probability that $A$ has repeated nonzero eigenvalues is zero. In other words,*

$$P_A(\{A \in \mathbb{R}^{n \times n} \mid \exists \lambda \in \mathbb{R} : \text{rank}(A - \lambda I) < n - 1\}) = P_A(\{A \in \mathbb{R}^{n \times n} \mid \text{rank}(A) < n - 1\})\,.$$

*Proof.* Let the random matrices $B, X$ that make up the sparse-continuous random matrix $A$ be defined on a probability space $(\Omega, A, P)$. For a fixed zero-pattern $S \subseteq [n]^2$ (specifying which entries are nonzero) we define

$$E_S := \{\omega \in \Omega \mid B_{ij}(\omega) = 1 \text{ iff } (i,j) \in S\} \cap \{\omega \in \Omega \mid A(\omega) \text{ has a repeated nonzero eigenvalue}\}\,.$$

We have $P(\{S(\omega) = S\}) = p^{|S|}(1-p)^{n^2-|S|}$ and conditionally on $S(\omega) = S$, $A(\omega)$ is a vector of $|S|$ continuous random variables $(X_{ij})_{(i,j)\in S}$ that is absolutely continuous with respect to the Lebesgue measure on $\mathbb{R}^{|S|}$. The matrix $A$ has a repeated nonzero eigenvalue if there exists an eigenvalue $\lambda \neq 0$ of algebraic multiplicity at least 2, i.e., $\det(A - \lambda I) = 0$ and $\frac{d}{d\lambda} \det(A - \lambda I) = 0$. These conditions define a nontrivial algebraic subset in $\mathbb{R}^{n \times n}$, imposing polynomial equations on these $|S|$ nonzero entries. These polynomials vanish only on a measure-zero subset of $\mathbb{R}^{|S|}$ (since the polynomials are not identically zero). Thus, $P(E_S) = 0$. And by the union bound,

$$P(\{A \text{ has a repeated nonzero eigenvalue}\}) \leq \sum_{S \subseteq [n]^2} P(E_S) = 0\,.$$

The second statement follows since the two events characterize having *some* repeated eigenvalue and *zero* being a repeated eigenvalue, respectively. $\square$

With these preliminaries, we find the following lower bound.

**Lemma 2.** *A sparse-continuous random matrix with sparsity $p$ is globally not identifiable with probability at least $1 - (1 - p^n)^n - np^n(1 - p^n)^{n-1}$ for $n \geq 2$ (and $p$ for $n = 1$).*

The proof we provide in Section B is driven by the presence of zero rows/columns. These correspond to sink or source nodes in the corresponding adjacency graph, which are typically frequent in sparse graphs and dominate the probability of rank deficiency. However, neither zero rows/columns nor an iid Bernoulli sparsity structure are required. In Lemma 5 we prove a non-zero lower bound for more general sparsity patterns that also allow for different degree distributions (e.g., accommodating hubs) and also empirically study a sparsity model that excludes zero rows/columns in Section D.6.

Next, we prove a sharp threshold on the dimension $n$ and sparsity level $p$ for global unidentifiability.

---

[1]The negation of "globally unidentifiable" is not called "globally identifiable" as this would convey the wrong impression of it being "always" identifiable, i.e., from any initial condition.

**Lemma 3** (sharp threshold for global unidentifiability). *Let $A$ be a sparse-continuous matrix with $n$-dependent sparsity level $p(n)$. Then, for $p_c(n) = 1 - \frac{\ln(n)}{n}$ and any function $\omega(n) \to \infty$ we have that if $p(n) = p_c(n) + \frac{\omega(n)}{n}$, then $P(\mathrm{rank}(A) \leq n - 2) \to 1$ for $n \to \infty$ and if $p(n) = p_c(n) - \frac{\omega(n)}{n}$, then $P(\mathrm{rank}(A) \leq n - 2) \to 0$ for $n \to \infty$. That is, there is a threshold at $p = \ln(n)/n$ decisive for whether $A$ is asymptotically globally unidentifiable with high probability or not.*

The proof can be found in Section B. This is consistent with known results on Bernoulli matrices (e.g., Frieze & Karoński, 2015; Basak & Rudelson, 2021) where there exists a similar sharp transition at $p = 1 - \ln(n)/n$ for the probability of singularity. Our proof extends the same threshold to two-fold rank deficiency in sparse-continuous matrices and thus to global unidentifiability. In practice, numerically evaluating matrix ranks requires thresholding. By the Eckart–Young–Mirsky theorem the second smallest singular value $\sigma_2$ of $A$ corresponds to the distance of $A$ the closest matrix $A'$ with $\mathrm{rank}(A') < n - 1$. As such it provides a robust continuous numerical measure of unidentifiability.

**Remark on sparsity, rank-deficiency, and non-identifiability.** Stanhope et al. (2014); Qiu et al. (2022) establish a tight connection between global non-identifiability and rank deficiency to then argue that rank deficiency (and thus unidentifiability) is rare. One of our key contributions is to demonstrate and quantify how sparsity, commonly assumed in dynamical models, leads to rank deficiency that ultimately results in unidentifiability.[2] Crucially, this link does not rely on sparsity inducing entire zero rows/columns (obviously giving rank deficiency), but also holds for sparse systems without zero rows/columns, see also Section D.6. Finally, we link to systems studied in the literature to corroborate that the found phenomena are relevant for real-world modeling Section C.

## 4 TRAJECTORY UNIDENTIFIABILITY

We now turn to "trajectory unidentifiability", i.e., the situation where $A$ is not globally unidentifiable, yet it may be unidentifiable from specific observed trajectories $\boldsymbol{x}$. Let $(A, \boldsymbol{x}_0) \sim P_A \otimes P_{\boldsymbol{x}_0}$ with an absolutely continuous $P_{\boldsymbol{x}_0}$ (w.r.t. the Lebesgue measure on $\mathbb{R}^n$) and $P_A$ the distribution of a sparse-continuous matrix $A$. Since, according to Theorem 2, $A$ is unidentifiable from $\boldsymbol{x}$ if and only if $\boldsymbol{x}_0$ is in a proper invariant subspace of $A$, and the fact that any *proper* subspace of $\mathbb{R}^n$ has zero probability under any absolutely continuous probability measure (with respect to the Lebesgue measure), we conclude that $P_{A,\boldsymbol{x}_0}(\mathcal{U}_A \mid A = A') = 0$ for all $A' \notin \mathcal{U}$. Therefore, the second term in Eq. (2) is zero. The probability of a "random" $A$ being identifiable from a "random" solution trajectory is thus given entirely by the probability of $A$ being globally unidentifiable—the probability of "unlucky" initial conditions is zero.

However, $\boldsymbol{x}_0$ being close to a proper $A$-invariant subspace can still be problematic in practice, which we later demonstrate empirically. To measure this "closeness to unidentifiability", we define

$$d_A : \mathbb{R}^n \to [0, 1], \quad \boldsymbol{x}_0 \mapsto \frac{1}{\|\boldsymbol{x}_0\|_2} \min\{\mathrm{dist}(\boldsymbol{x}_0, V) \mid V \in \mathcal{I}(A)\}, \tag{3}$$

where $\mathcal{I}(A)$ is the set of proper $A$-invariant linear subspaces of $\mathbb{R}^n$ and $\mathrm{dist}(\boldsymbol{x}_0, V) := \min_{\boldsymbol{y} \in V} \|\boldsymbol{x}_0 - \boldsymbol{y}\|_2$ is the Euclidean distance of $\boldsymbol{x}_0$ from the subspace $V$. We have that $d_A(\boldsymbol{x}_0) = 0$ if and only if $\boldsymbol{x}_0$ lies in a proper $A$-invariant subspace, and $d_A$ provides a continuous measure for the "level of unidentifiability from the trajectory," formalized as follows.

**Lemma 4.** *Let $A, A' \in \mathbb{R}^{n \times n}$ and assume there is an $A$-invariant subspace $V^*$ such that $(A - A')V^* = \{0\}$. Then, for every $t \geq 0$ and $\boldsymbol{x}_0 \in \mathbb{R}^n$, we have that*

$$\|e^{At}\boldsymbol{x}_0 - e^{A't}\boldsymbol{x}_0\|_2 \leq C(t, A, A')\|A - A'\|_2 \, d_A(\boldsymbol{x}_0)$$

*with $C(t, A, A') := \int_0^t \|e^{A(t-s)}\|_2 \|e^{A's}\|_2 ds$. Further, for any $\varepsilon > 0$, the condition*

$$\|(e^{At} - e^{A't})\,\boldsymbol{x}_0\| \leq \varepsilon \quad \text{for all } 0 \leq t \leq T$$

*holds whenever*

$$T \leq \frac{1}{\alpha} W\left(\alpha \frac{\varepsilon}{\|A - A'\| M^2 \, d_A(\boldsymbol{x}_0)}\right),$$

*where $W(\cdot)$ denotes the Lambert function and constants $\alpha \in \mathbb{R}, M \geq 1$ such that $\|e^{At}\|, \|e^{A't}\| \leq Me^{\alpha t}$ for all $t \geq 0$.*

---

[2]Generally sparse matrices can have full rank (e.g., identity) and full rank matrices can be dense (all ones).

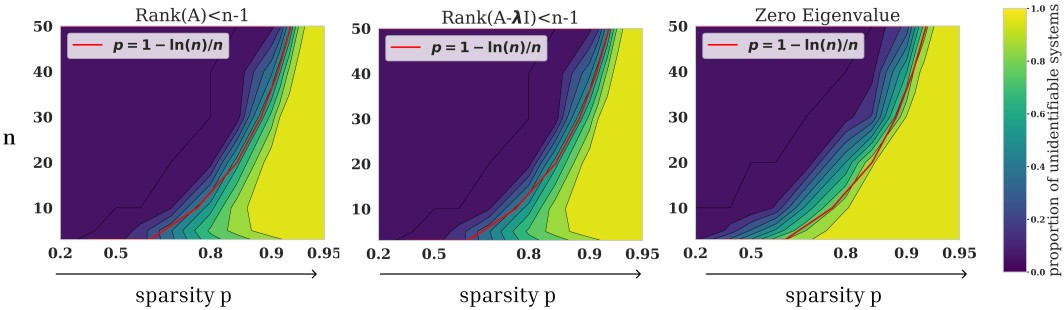

Figure 2: Proportion of matrices satisfying the conditions i), ii) and iii) at different system dimensions $n$ and sparsity levels $p$. All three conditions exhibit a sharp increase in frequency with higher sparsity, consistent with the threshold $p = 1 - \ln(n)/n$.

The proof can be found in Section B. Assume $A$ is identifiable from $\boldsymbol{x}_0$, but $\boldsymbol{x}_0$ is close to a proper $A$-invariant subspace $V^*$ with distance $d_A(\boldsymbol{x}_0)$. Then there exists another $A' \neq A$ that agrees with $A$ on $V^*$, i.e., $A'$ could practically be mistaken for the underlying true dynamic. Lemma 4 guarantees $\epsilon$-closeness of $A$ and $A'$ up to time $T$, which scales inversely with both the distance $d_A(\boldsymbol{x}_0)$ and the norm $\|A - A'\|_2$, so trajectories remain indistinguishable for longer horizons when the initial condition lies closer to an invariant subspace or when competing matrices $A$ and $A'$ differ only slightly. While lemma 4 holds for general matrices, it is particularly relevant for sparse matrices as these are more likely to have large invariant subspaces and are consequently more likely unidentifiable in practice.

## 5 EMPIRICAL RESULTS

Our empirical validation experiments include three distinct viewpoints: firstly, we assume the true system matrix to be available in order to confirm our theoretical results on unidentifiability in sparse matrices in principle. Secondly, we assume that we only have observed trajectories at hand and evaluate trajectory-level identifiability criteria. While these results assess whether a system is (or is not) identifiable, we finally also empirically assess the performance of two widely used estimators that attempt to learn the underlying system from data directly, hence zooming in on identifiability challenges in practice.

**Data generation.** We focus on systems of dimensions $n \in \mathcal{D} := \{3, 5, 10, 20, 30, 40, 50\}$ and sparsity levels $p \in \mathcal{P} = \{0.1, 0.2, 0.3, 0.4, 0.5, 0.6, 0.7, 0.8, 0.9, 0.95, 1.0\}$.[3] For each $(n, p) \in \mathcal{D} \times \mathcal{P}$, we generate 100 matrices according to the sparse-continuous model in Definition 3 and take a standard normal for the continuous distribution $P_X = \mathcal{N}(0, 1)$. For each matrix, we sample 100 initial conditions uniformly from the unit circle in $\mathbb{R}^n$ and solve the corresponding trajectories numerically over the interval $t \in [0, 1]$ using a standard explicit RK45 solver with 512 homogeneous time steps. Results for different random matrix models, for example disallowing zero rows/columns, can be found in Section D.

### 5.1 SYSTEM-LEVEL UNIDENTIFIABILITY

In this section, we perform an exploratory analysis of global unidentifiability conditions by examining properties at the system level. The contour map in Fig. 2 illustrates, for varying values of sparsity $p$ and dimension $n$, the empirical frequency with which the generated system matrices $A$ violate at least one of three structural criteria: (i) $\mathrm{rank}(A) < n - 1$; (ii) there exists an eigenvalue $\lambda$ for which $\mathrm{rank}(A - \lambda I) < n - 1$; or (iii) the spectrum contains the zero eigenvalue. As matrices become sparser (i.e., as $p$ increases), the proportion of globally unidentifiable matrices sharply increases. Notably, the nearly identical contour plots for conditions (i) and (ii) strongly support our theoretical result stated in Lemma 1. Moreover, the condition regarding the existence of zero eigenvalues aligns closely with the theoretical threshold $p = 1 - \frac{\ln(n)}{n}$ (depicted by the red curve).

**Discussion on realistic sparsity levels.** Although global (system-level) unidentifiability becomes less critical as the dimension increases (consistent with the threshold $p = 1 - \frac{\ln(n)}{n}$), seemingly

---

[3]Recall that higher $p$ for us corresponds to "more zeros."

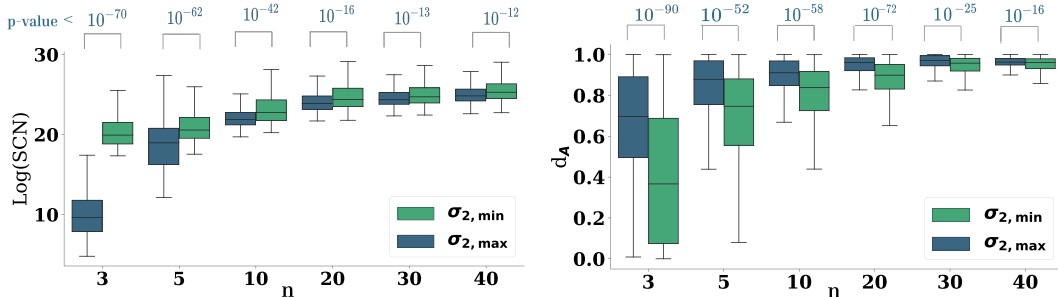

Figure 3: Box-plots of smoothed condition numbers (SCN) and distance-to-unidentifiability $d_A$ for the least and most identifiable groups of systems at different dimensions $n$.

only leaving systems of negligibly high sparsity-levels unidentifiable, we emphasize that such high sparsity regimes are not merely theoretical but are indeed realistic and frequently encountered in real-world scenarios. For instance, the *P. trichocarpa* PEN gold standard network (Walker et al., 2022), introduced earlier, has a sparsity level of approximately $0.997$ for a graph comprising $1690$ nodes (genes), exceeding the theoretical threshold $p^* = 1 - \frac{\ln(n)}{n} \approx 0.996$. Similarly, the *E. coli* gene regulatory network (Walker et al., 2022), consisting of $1222$ genes, exhibits a sparsity of about $0.999$, surpassing its corresponding threshold of $p^* = 0.995$.

## 5.2 TRAJECTORY-LEVEL UNIDENTIFIABILITY

Following the argument in Section 3, the second-smallest singular value $\sigma_2$ of a system relates to its global unidentifiability. Here, we ask whether systems with worse global unidentifiability metrics also give rise to trajectories that are statistically less identifiable. To address this question, we rank the generated system matrices by their second-smallest singular value $\sigma_2$ for each combination of dimension $n$ and sparsity $p$. Matrices with $\sigma_2$ falling into the top-10% of ranked singular values are subsequently referred to as $A_{\sigma_2,max}$; similarly, matrices at the bottom-10% are referred to as $A_{\sigma_2,min}$.

To determine whether the trajectories produced by these two matrix groups differ in identifiability, we compared the sample means of two complementary trajectory-level identifiability metrics: our closeness-to-unidentifiability distance $d_{A,0}(x_0)$ from Section 4 (larger implies more likely identifiable) as well as the smoothed condition number (SCN) as established by Qiu et al. (2022) (smaller implies more likely identifiable). Specifically, we approximate $d_A(x_0)$ by the normalized distance of the initial state to $\ker(A)$, using the metric

$$d_A(\boldsymbol{x}_0) \approx d_{A,0}(\boldsymbol{x}_0) := \frac{1}{\|x_0\|} \text{dist}(\boldsymbol{x}_0, \ker(A)) \in [0,1]. \tag{4}$$

Compared to $d_A(x_0)$, $d_{A,0}(x_0)$ allows us to estimate "distance-to-unidentifiability" without analyzing all proper $A$-invariant subspaces. As a second criterion, we evaluate the invertibility of the pairwise inner product matrix, as established by Qiu et al. (2022), who showed that a trajectory $\boldsymbol{x}(t \mid A, \boldsymbol{x}_0)$ is identifiable if and only if the pairwise inner product matrix $\Sigma_{xx} = \langle \boldsymbol{x}(t), \boldsymbol{x}(t) \rangle_{i,j} = \int_0^T x_i(t)x_j(t)dt$ is invertible. Qiu et al. (2022) offer a straightforward method for approximating $\Sigma_{xx}$, and introduce the Smoothed Condition Number (SCN) $k(\Sigma_{xx})$ as the condition number of $\Sigma_{xx}$.

**Results.** Boxplots of SCN and $d_A$ values are depicted in Fig. 3 at different dimensions $n$. For each $n$ we focus on the sparsity parameter $p^*$ closest to the threshold $1 - \ln(n)/n$, which corresponds to the critical sparsity value at which a system becomes unidentifiable. We use Welch's t-test with one-sided alternatives that reflect the expected direction of the difference:

$$\text{SCN:} \qquad H_0 : \mu^{SCN}_{A_{\sigma_2,max}} \geq \mu^{SCN}_{A_{\sigma_2,min}} \qquad H_1 : \mu^{SCN}_{A_{\sigma_2,max}} < \mu^{SCN}_{A_{\sigma_2,min}}$$

$$d_{A,0}(x_0): \qquad H_0 : \mu^{d_A}_{A_{\sigma_2,min}} \geq \mu^{d_A}_{A_{\sigma_2,max}} \qquad H_1 : \mu^{d_A}_{A_{\sigma_2,min}} < \mu^{d_A}_{A_{\sigma_2,max}}$$

Here $\mu_{A_{\sigma_2,max}}$ and $\mu_{A_{\sigma_2,min}}$ denote the population means in the two subgroups. The goal of the statistical test is to establish whether the two subgroups truly differ in trajectory-identifiability performance. We find that systems in subgroup $A_{\sigma_2,min}$ are significantly less probable to identify ($p \ll 0.01$) for all system dimensions and both criteria. These results are consistent with our theoretical derivations and also in line with the previously analyzed systems-level criteria

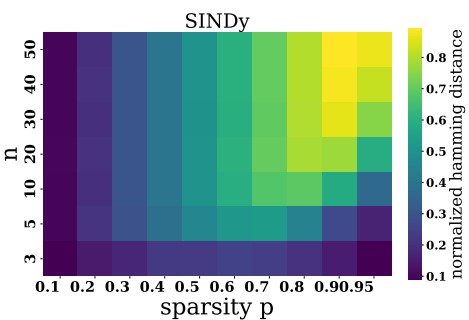 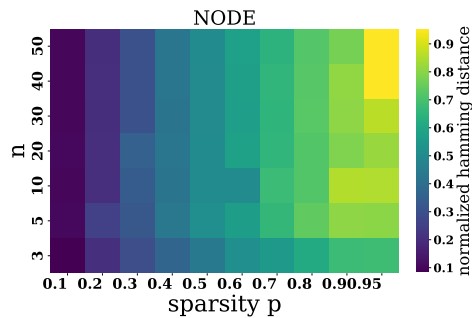

Figure 4: Normalized Hamming distance of reconstructed systems using SINDy (left) and NODE (right) across varying dimensions and sparsity levels. Reconstruction accuracy declines noticeably with increasing dimensionality and sparsity.

### 5.3 EMPIRICAL UNIDENTIFIABILITY

We evaluate the performance of two state-of-the-art models for identifying dynamical systems: Neural ODEs (**NODE**s) (Chen et al., 2018), which approximate dynamics through a neural-network-based function $f_\theta(\boldsymbol{x}(t))$, as well as the Sparse Identification of Nonlinear Dynamics algorithm (**SINDy**) (Brunton et al., 2016), which employs $L_2$-regularized linear regression on a predefined set of basis functions. To effectively model sparse linear systems, we utilize an $L_1$–regularized neural network without activation functions for NODEs, and linear basis functions for SINDy. Hyper-parameters for both models are tuned per system. More details on the methods can be found in Section D.3.

Since we are less interested in the question of whether the model can fit the observed data overall, but rather in the question of whether the models correctly identify the underlying system matrices $A$ in cases where they do fit the data well, we filter out estimates for which the corresponding reconstruction of the observed trajectory does not satisfy prescribed $R^2$ and MSE thresholds, where these metrics compare observed and reconstructed trajectories. This best-case filtering ensures that subsequent empirical results can be tied to identifiability properties rather than potential issues with model optimization or architecture.

**Results.** As perfect point estimates of all coefficients of system matrix $A$ are unreasonable to expect (e.g. due to numerical issues), we instead use the Hamming distance to compare binarized ground truth $A$ and estimate $\hat{A}$, which we additionally normalize by $n^2$ for comparability across dimensions. The Hamming distance corresponds to the number of matching symbols (here: zero or one) and thus captures the overlap of the predicted and true sparsity patterns. The heatmaps in Fig. 4 show for both SINDy and NODE a clear left-right gradient, indicating that sparse systems lead to a larger Hamming distance - consistent with our theoretical finding that higher sparsity increases the probability of unidentifiability. In addition, results for SINDy also show an unexpected up-down gradient and high sparsity levels, which were not predicted by our theory. This behavior can be tied to the (default) sequentially thresholded least-squared optimizer that is used to fit SINDy to the data: Following least-squared optimization, this optimizer sets any coefficient that falls below a user-defined magnitude threshold to zero - hence promoting sparsity aggressively in the fitted model. In cases of very high sparsity and low system dimensionality, most coefficients of the true system matrix will be zero so that thresholding coefficients towards zero biases the model towards a smaller Hamming distances. As the dimensionality increases or the sparsity reduces towards moderate sparsity levels, this effect vanishes as there are multiple non-zero coefficients.

Exemplary model estimates and corresponding trajectory fits are displayed for a selected dense and sparse system in Fig. 1, which further illustrates the phenomenon.

## 6 CONCLUSION

In this work, we focus on identifiability of sparse linear ODEs $A$ from a single observed trajectory $\boldsymbol{x}$. We first partition the probability of unidentifiability of $A$ from $\boldsymbol{x}$ into "system level unidentifiability" ($A \in \mathcal{U}$) and "trajectory level unidentifiability", i.e., hitting an "unlucky" initial condition. After showing that the latter scenario almost surely does not happen, we show that in sparse systems the probability of unidentifiability can be lower bounded by a positive quantity and exhibits a sharp

asymptotic threshold at $p = 1 - \ln(n)/n$ for global unidentifiability. This puts an important asterisk on the celebrated result that "almost all linear ODEs are identifiable," which ceases to hold under sparsity, which is often a core underlying assumption in dynamical system modeling. We go on to empirically verify that theoretical unidentifiability is a serious practical challenge and extend our analysis to near-unidentifiability on the "trajectory level," demonstrating that the theoretically negligible case of unlucky initial conditions further exacerbates the problem. Important directions for future work include extending our results to affine and post-nonlinear models, discrete, noisy or partially observed systems, multiple observed trajectories, and deriving computable metrics for practical unidentifiability from data alone.

## REPRODUCIBILITY STATEMENT

Significant effort was made to ensure reproducibility, both for the theoretical and experimental results. The complete proofs of Lemma 2, Lemma 3, Lemma 4 can be found in Section B.1, Section B.2, Section B.3 respectively. Additionally, we add a discussion on the assumptions and limitations in Section 2. Details regarding implementation and metrics can be found in Section D, specifically software resources in Section D.1, metrics details in Section D.2 and methods implementation details in Section D.3. The data generating mechanism is described in Section 5. Code is publicly available at https://github.com/ceciliacasolo/ode-identifiability.

## THE USE OF LARGE LANGUAGE MODELS

In this work, Large Language Models (LLMs) were used as a writing assistant to polish the text and improve clarity of exposition.

## ACKNOWLEDGMENTS

This work is supported by the DAAD programme Konrad Zuse Schools of Excellence in Artificial Intelligence, sponsored by the Federal Ministry of Education and Research. This work has been supported by the German Federal Ministry of Education and Research (Grant: 01IS24082). The authors gratefully acknowledge the Gauss Centre for Supercomputing e.V. (www.gauss-centre.eu) for funding this project by providing computing time through the John von Neumann Institute for Computing (NIC) on the GCS Supercomputer JUWELS at Jülich Supercomputing Centre (JSC).

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

# A    DISCUSSION OF OUR ASSUMPTIONS AND THEIR IMPLICATIONS

For our theoretical work we assume continuous, noise-free, fully-observed trajectories, which may at first glance appear limiting. Indeed, those assumptions would be limiting if this was a method proposal, where the method only works on such observations. However, for analyzing unidentifiability, they are the right starting point: Without these assumptions, no system would ever be identifiable, i.e., unidentifiability would be trivially given. Only once we have eliminated unidentifiability due to, e.g., undersampling, finite samples, and partial observations, can we make non-trivial claims about the inherent unidentifiability of the underlying systems. We split the discussion of our assumptions into two parts.

1. The assumptions of *continuous* trajectories, *noise-free* observations and *full observability* are assumptions about our ability to observe/measure the system. For these, our setting corresponds to the "best case" idealized setting where (in principle) infinite data (both in terms of sampling frequency as well as in terms of noise instantiations) are available about all relevant parts of the system. As we will discuss one-by-one below, violating any of those assumptions will add additional identifiability issues, which are due to limitations of our observation process (not inherent to the system) that one could in principle overcome. Hence, instead of "our results only apply under those assumptions," our results show unidentifiability "despite these assumptions", even when all solvable sources of unidentifiability due to limited measurement processes have been overcome. In short, these form the appropriate starting point for an identifiability analysis.

2. The assumption of linear systems and sparsity are indeed assumptions about the underlying dynamics. We elaborate on the relevance of these assumptions below.

**1a. Continuous trajectories.**    Any real-world measurement device will have a maximum sampling frequency, so all real-world data is necessarily discrete in time. Discrete observations add trivial unidentifiability issues around aliasing (cf. aliasing in the Nyquist sampling theorem). Hence, the continuous observation case is the best-case scenario - the limit of infinite sampling frequency - where we expect any remaining unidentifiability to be due to the fundamental problem setting, not our insufficient sampling rate.

**1b. Noise-free observations.**    Again, real-world data are almost always noisy. And again, in the presence of noise, system identification becomes inherently more difficult, namely probabilistic: multiple different systems may be compatible with the observed data, each with a different likelihood under the assumed noise model (for example, the standard assumption of i.i.d. additive Gaussian noise in classical machine learning often used to model "measurement errors"). Within this probabilistic framework, our noise-free setting can be interpreted as the best case scenario, where unidentifiability is not just due to suboptimal measurement devices. It can also be interpreted as the asymptotic limit of observing infinitely many trajectories for a single initial condition such that the "noise can be averaged out" to recover the noise-free setting. From this perspective, our results state under which conditions collecting more observations allows (in the asymptotic limit) the identification of the system. Or put differently, we show that there are cases in sparse systems where, even if more data was collected or where the noise level is reduced by other means, the true system remains unidentifiable.

**1c. Full observations.**    Akin to discrete or noisy observations, only having partial observations of the system state will naturally render identifiability more difficult. Unlike discrete observations and noise, partial observations add a fundamental degree of unidentifiability as there is no "asymptotic limit" of more frequent or precise observations to recover our setting. Intuitively, when we allow for the presence of (an arbitrary number of) unobserved state variables, both the full system as well as the "observed part" (viewed as a submatrix of the full system) are inherently unidentifiable. As an extreme (and somewhat pathological) counterexample, one could imagine an exact copy of each observed variable among the latents such that all observed variables may either depend on the unobserved copy or the observed variable, which is indistinguishable from data. Hence, to study the identifiability under partial observations, one would have to provide additional limiting assumptions on the allowed type and nature of confounding. Exploring the palette of possible assumptions is an interesting direction for future work.

**2a. Linear systems.**    There are two main reasons for this assumption: (a) Linear differential equation models are a core pillar of scientific modeling and widely used in practice due to their analytical

tractability, relative simplicity, and inherent interpretability. While the real underlying dynamics of complex systems are arguably rarely perfectly linear, linear models also turn out to be surprisingly useful and predictive (akin to linear regression still being a competitive and reliable tool in many settings). (b) Non-linear ODEs are known to be unidentifiable from single trajectories if one does not impose heavy constraints on the allowed non-linear function family (Scholl et al., 2023). Hence, the previous proclaimed state in the literature was a clear separation: "non-linear systems are unidentifiable while linear systems are almost surely identifiable." Our work adds important nuances to the part about linear systems.

**2b. Sparsity.** If it were somehow known that real-world systems never evolve according to dynamics below the sparsity-identifiability-threshold, our findings would indeed be of purely theoretical interest. However, the true sparsity of real-world dynamics remains inherently unknown. While it is impossible to determine whether a given unknown system is sparse, the examples in our submission indicate that this may not be unlikely in practice. Our results are thus important in that modeling practitioners should consider them when interpreting their results, e.g., despite a perfectly-fit model, the inferred weights might not reflect the true causal relations between variables due to unidentifiability issues. As a pragmatic step to highlight that high degrees of sparsity may occur in practically relevant settings, we examined published sparse systems, which we discuss in 5.1. In both reported cases, the published systems are on the sparser side of the threshold required for identifiability, which further supports the practical relevance of our findings. Nonetheless, we emphasize that the relevance of our results in terms of how frequently they show up cannot be empirically proven, since the notion of sparsity in our analysis refers to that of the true but unknown dynamics, rather than the model dynamics (models cannot be proven "true"). However, a practical implication for future method development efforts aimed at inferring the "ground truth" networks is for instance that such methods may produce estimates that fit the observed data equally well as the ground truth, yet are different. Importantly, this discrepancy might not stem from the limitations of the developed method itself, but rather from the nature of the underlying dynamics being estimated.

# B    THEORETICAL RESULTS

## B.1    PROOF OF LEMMA 2

**Lemma 2.** *A sparse-continuous random matrix with sparsity $p$ is globally not identifiable with probability at least $1 - (1 - p^n)^n - np^n(1 - p^n)^{n-1}$ for $n \geq 2$ (and $p$ for $n = 1$).*

*Proof.* Following Theorem 1, the model Eq. (1) is globally unidentifiable if and only if $A$ has more than one Jordan normal block for one of its eigenvalues, say $\lambda_i$, which then has geometric multiplicity $g(\lambda_i) > 1$, i.e., $\operatorname{rank}(A - \lambda_i I) < n - 1$. According to Lemma 1, it follows that $P_A(\mathcal{U}) = P_A(\{A \in \mathbb{R}^{n \times n} \mid \operatorname{rank}(A) < n - 1\})$. Hence, we are looking for the probability of dependencies among columns in $A$ that drop the rank. For a fixed zero structure $B := (b_{i,j})_{i,j \in [n]} \in \{0, 1\}^{n \times n}$ the set on which $X := (x_{i,j})_{i,j \in [n]} \in \mathbb{R}^{n \times n}$ introduces additional linear dependencies among the rows/columns of $A$ has Lebesgue measure zero and thus zero probability under $A$. Therefore, $P(\operatorname{rank}(A) < n - 1) = P(\operatorname{rank}(B) < n - 1)$. We focus on the sufficient event that $B$ has multiple zero columns. Since the entries of $B$ are independent, $P(B_{:,i} = 0) = p^n$ for all $i \in [n]$ where $B_{:,i}$ represents the $i$-th column, it follows that the random variable $Z$ representing the count of zero columns follows a Binomial distribution $Z \sim \operatorname{Bin}(n, q)$ with $q := p^n$. With the probability mass function $P(Z = k) = \binom{n}{k} q^k (1 - q)^{n-k}$ we find

$$P(\operatorname{rank}(A) < n - 1) \geq P(Z \geq 2) = 1 - P(Z \in \{0, 1\}) = 1 - (1 - p^n)^n - np^n(1 - p^n)^{n-1} .$$

$\square$

Lemma 2 provides a non-zero lower bound on the probability of system level unidentifiability for the simple iid Bernoulli sparsity pattern. However, by analogous reasoning the proof extends to a much broader class of structured sparsity patterns that also allow for the existence of hubs, i.e., for heavy tailed degree distributions.

**Lemma 5** (General sparsity-pattern lower bound). *Let $P_B$ be a probability measure on $\{0, 1\}^{n \times n}$ and let $B \sim P_B$. Consider a sparse-continuous random matrix $A := (B_{ij} X_{ij})_{i,j=1}^n$, where $(X_{ij})_{i,j=1}^n$*

*are independent continuous random variables (i.e., their joint distribution is absolutely continuous with respect to the Lebesgue measure on $\mathbb{R}^{n \times n}$) and independent of B. Assume that for some $k \in \{0, 1, \ldots, n-2\}$ there exists a set of columns $J \subseteq [n]$ with $|J| \geq k+2$ and a set of rows $R \subseteq [n]$ with $|R| \leq k$ such that the event*

$$E_k := \left\{ B \in \{0,1\}^{n \times n} \;\middle|\; B_{ij} = 0 \text{ for all } j \in J, \, i \notin R \right\} \tag{5}$$

*has positive probability $q := P_B(E_k) > 0$. Then A is globally unidentifiable with probability at least $q$, i.e., $P(A \in \mathcal{U}) \geq q > 0$.*

*Proof.* The key idea is that every deterministic matrix $M \in \mathbb{R}^{n \times n}$ whose zero pattern belongs to $E_k$ has $\operatorname{rank}(M) \leq n-2$ and is therefore system level unidentifiable. For such matrices, for every $j \in J$ the $j$-th column $M_{:,j}$ is supported only on rows in $R$. Hence each $M_{:,j}$ with $j \in J$ lies in an $|R|$-dimensional subspace. We thus found at least $k+2$ columns of $M$ that lie in the same $|R| \leq k$ dimensional subspace, which decreases the rank of $M$ by at least 2 and by Theorem 1, $M$ is globally unidentifiable. Now consider the random matrix $A = (B_{ij} X_{ij})$. Since the above argument only depends on $B$, we conclude that $\operatorname{rank}(A) \leq n-2$ whenever $B \in E_k$. Hence $P(A \in \mathcal{U} \mid B) = 1$ for all $B \in E_k$ and by total probability

$$P(A \in \mathcal{U}) = \mathbb{E}_B\big[P(A \in \mathcal{U} \mid B)\big] \geq \mathbb{E}_B\big[\chi_{E_k}(B)\, P(A \in \mathcal{U} \mid B)\big] = \mathbb{P}_B(E_k) = q > 0 \,.$$

$\square$

The main takeaway is that as long as there may be some variables that are sparsely connected among each other with a sufficient degree of sparsity there will be a non-zero probability of system level identifiability—in contrast to the dense setting.

## B.2 PROOF OF LEMMA 3

**Lemma 3** (sharp threshold for global unidentifiability). *Let A be a sparse-continuous matrix with $n$-dependent sparsity level $p(n)$. Then, for $p_c(n) = 1 - \frac{\ln(n)}{n}$ and any function $\omega(n) \to \infty$ we have that if $p(n) = p_c(n) + \frac{\omega(n)}{n}$, then $P(\operatorname{rank}(A) \leq n-2) \to 1$ for $n \to \infty$ and if $p(n) = p_c(n) - \frac{\omega(n)}{n}$, then $P(\operatorname{rank}(A) \leq n-2) \to 0$ for $n \to \infty$. That is, there is a threshold at $p = \ln(n)/n$ decisive for whether A is asymptotically globally unidentifiable with high probability or not.*

*Proof.* From $A$ form the random bipartite graph $G_{n,n,s}$ with $A$ as the corresponding adjacency matrix. Edges are present independently with probability $s = s(n) = 1 - p(n)$.

For every bipartite graph $G_{n,n,s}$ let

$$m(G) = \max\{|M| : M \text{ is a matching}\}, \qquad d(G) = \max_{S \subseteq [n]} \big(|S| - |N(S)|\big) \tag{6}$$

(Hall deficiency). From Hall-König's theory for graph matching, the rank of the adjacency matrix $A$

$$m(G) \; \leq \; \operatorname{rank} A \; \leq \; n - d(G). \tag{7}$$

Given

$$s(n) = \frac{\ln n + \alpha(n)}{n},$$

we have that (Frieze & Karoński, 2015) $P(m(G) = n) \to 1$ if $\alpha(n) \to \infty$ and $P(m(G) = n) \to 0$ if $\alpha(n) \to -\infty$.

*Case $\alpha(n) \to -\infty$.* Let $Z$ be the number of isolated vertices in $G_{n,n,s}$. Without loss of generalization, we will consider isolated vertices from rows. Any fixed vertex is isolated with probability $(1-s)^n$ so

$$Z \sim \operatorname{Bin}\big(n, (1-s)^n\big), \qquad \mathbb{E}[Z] = n(1-s)^n = e^{-\alpha(n)}(1+o(1)), \qquad \operatorname{Var}[Z] = O(\mathbb{E}[Z]).$$

Chebyshev's inequality therefore yields $P(Z \geq 2) \to 1$ for $n \to \infty$. Two isolated vertices form a set $S$ with $|N(S)| \leq |S| - 2$, so $d(G) \geq 2$ and consequently $\Pr[\operatorname{rank} A \leq n-2] \to 1$.

*Case $\alpha(n) \to +\infty$.* Then $P(m(G) = n) \to 1$, hence $P(\operatorname{rank}(A) \leq n-2) \to 0$. Substituting $s = 1 - p$ concludes the proof. $\square$

### B.3 PROOF OF LEMMA 4

**Lemma 4.** *Let $A, A' \in \mathbb{R}^{n \times n}$ and assume there is an $A$-invariant subspace $V^*$ such that $(A - A')V^* = \{0\}$. Then, for every $t \geq 0$ and $\boldsymbol{x}_0 \in \mathbb{R}^n$, we have that*

$$\|e^{At}\boldsymbol{x}_0 - e^{A't}\boldsymbol{x}_0\|_2 \leq C(t, A, A')\|A - A'\|_2 \, d_A(\boldsymbol{x}_0)$$

*with $C(t, A, A') := \int_0^t \|e^{A(t-s)}\|_2 \|e^{A's}\|_2 ds$. Further, for any $\varepsilon > 0$, the condition*

$$\|(e^{At} - e^{A't})\boldsymbol{x}_0\| \leq \varepsilon \quad \text{for all } 0 \leq t \leq T$$

*holds whenever*

$$T \leq \frac{1}{\alpha} W\Big(\alpha \frac{\varepsilon}{\|A - A'\| \, M^2 \, d_A(\boldsymbol{x}_0)}\Big),$$

*where $W(\cdot)$ denotes the Lambert function and constants $\alpha \in \mathbb{R}, M \geq 1$ such that $\|e^{At}\|, \|e^{A't}\| \leq Me^{\alpha t}$ for all $t \geq 0$.*

*Proof.* Let us assume without loss of generality (otherwise we simply get a loser bound) that $V^*$ is the closest proper $A$-invariant subspace to $\boldsymbol{x}_0$. We can then decompose $\boldsymbol{x}_0$ as $\boldsymbol{x}_0 = \Pi_{V^*}\boldsymbol{x}_0 + \boldsymbol{w}$ with $\|\boldsymbol{w}\| = d_A(\boldsymbol{x}_0)$ (by definition, all norms are 2-norms). Given the assumption on $A$ and $A'$, we have

$$(e^{At} - e^{A't})\boldsymbol{x}_0 = (e^{At} - e^{A't})(\Pi_{V^*}\boldsymbol{x}_0 + \boldsymbol{w}) = (e^{At} - e^{A't})\boldsymbol{w} \, .$$

From a variation of constants approach and the triangle inequality, we have

$$\|e^{At} - e^{A't}\| = \Big\| \int_0^t e^{A(t-s)}(A - A')e^{A's}ds \Big\| \leq \int_0^t \|e^{A(t-s)}\| \|e^{A's}\| \|A - A'\| ds \, ,$$

which ultimately gives the result

$$\|e^{At}\boldsymbol{x}_0 - e^{A't}\boldsymbol{x}_0\| \leq \|e^{At} - e^{A't}\| \|\boldsymbol{w}\| \leq \|A - A'\| \, d_A(\boldsymbol{x}_0) \int_0^t \|e^{A(t-s)}\| \|e^{A's}\| \, ds \, .$$

For the second statement, we note that

$$\|(e^{At} - e^{A't})\boldsymbol{w}\| \leq \|A - A'\| \, M^2 \, \|\boldsymbol{w}\| \left( \int_0^t e^{\alpha t} \, ds \right) = \|A - A'\| \, M^2 \, d_A(\boldsymbol{x}_0) \, t \, e^{\alpha t} \, ,$$

which is bounded by $\varepsilon$ for $t \in [0, T]$ with

$$Te^{\alpha T} = \frac{\varepsilon}{\|A - A'\| \, M^2 \, d_A(\boldsymbol{x}_0)} \, .$$

Using the definition of the Lambert W function $W(\cdot)$ we obtain

$$T = \frac{1}{\alpha} W\Big(\alpha \frac{\varepsilon}{\|A - A'\| \, M^2 \, d_A(\boldsymbol{x}_0)}\Big) \, .$$

$\square$

## C EMPIRICAL EVIDENCE OF NON-IDENTIFIABILITY IN GRNS

We elaborate on the practical relevance of non-identifiability by discussing real-world gene regulatory networks (GRNs) that may exhibit structural features leading to non-identifiable dynamics. To illustrate this, we examined the gold-standard *E. coli* network from Marbach et al. (2012) (Supplementary Data 1). The adjacency matrix derived from this network contains several zero columns, indicating genes with no outgoing regulatory edges. This can be checked by analysing the `DREAM5_NetworkInference_GoldStandard_Network3.tsv` data. As discussed in our main text, zero columns in the adjacency matrix are a key source of non-identifiability. A second example provided by Marbach et al. (2012), the *S. cerevisiae* network (Network 4), also contains zero columns and thus similar issues.

For some other networks, such as *P. trichocarpa* and *A. thaliana*, we were not able to obtain the full gold-standard network, and so a definitive identifiability analysis is not possible. However, we

Table 1: Summary of real-world gene regulatory networks and their sparsity levels. The sparsity threshold corresponds to $1 - \frac{\log n}{n}$ from Lemma 3.

| Name | # Nodes ($n$) | # Edges | Sparsity ($p$) | Threshold ($1 - \frac{\log n}{n}$) |
|------|------|------|------|------|
| *A. thaliana* | 2,864 | 18,663 | 0.9977 | 0.9972 |
| *P. trichocarpa* | 1,690 | 9,268 | 0.9968 | 0.9957 |

summarize in Table 1 the available statistics from Walker et al. (2022), including the number of nodes, number of edges, sparsity level, and the theoretical sparsity threshold from Lemma 2 of our work.

In both cases, the sparsity level lies below the identifiability threshold, suggesting a high probability of non-identifiability. More broadly, the key point is that identifiability cannot be assessed purely from data. For any given system, even if it is theoretically identifiable, one cannot determine from observed data alone whether the inferred model corresponds to the ground truth. When fitting a linear ODE model to data from a sparse system, we may recover a model that exactly reproduces observations, yet it may be structurally incorrect. Thus, regardless of whether the true system is globally identifiable, it remains unidentifiable from data alone.

Moreover, the datasets discussed above should not be viewed as perfect ground truths. As noted by Walker et al. (2022, see §2.5), these "gold standard" networks are incomplete and potentially inaccurate. As such, both the exact sparsity patterns (*E. coli*) and sparsity levels (*P. trichocarpa*) may be unreliable, and conclusions about identifiability must be treated cautiously.

## D   EXPERIMENTAL DETAILS

### D.1   SOFTWARE

We provide the resources with the corresponding licenses used in this work in Table 2.

Table 2: Overview of resources used in our work.

| Name | Reference | License |
|------|------|------|
| Python | (van Rossum & Drake, 2009) | PSF License |
| PyTorch | (Paszke et al., 2019) | BSD-style license |
| Numpy | (Harris et al., 2020) | BSD-style license |
| Pandas | (pandas development team, 2020; Wes McKinney, 2010) | BSD-style license |
| Matplotlib | (Hunter, 2007) | modified PSF (BSD compatible) |
| Scikit-learn | (Pedregosa et al., 2011) | BSD 3-Clause |
| SciPy | (Virtanen et al., 2020) | BSD 3-Clause |
| SLURM | (Yoo et al., 2003) | modified GNU GPL v2 |
| networkx | (Hagberg et al., 2008) | BSD 3-Clause |
| JAX | (Bradbury et al., 2018) | Apache-2.0 |

### D.2   METRICS

**System-level identifiability metrics.**  To compute system level identifiability metrics, we perform a batched singular-value decomposition on every system matrix $A$ using `jax.numpy.linalg.svd`. Subsequently, any singular value $\sigma$ with $|\sigma| < 10^{-6}$ is treated as numerically zero. Eigenvalues are computed with `jax.numpy.linalg.eigvals` and the matrix rank is computed via `jax.numpy.linalg.matrix_rank` with tolerance level set to $10^{-6}$.

**Trajectory-level identifiability metrics.**  Smoothed condition number (SCN) analysis begins by constructing the empirical Gram matrix $\hat{\Sigma}_{xx} = YSY^T \in \mathbb{R}^{d \times d}$, where $Y = [x(t_1) \ldots x(t_n)]$ collects one simulated trajectory and where the diagonal "smoothing" matrix $S$ contains the numerical quadrature weights (trapezoidal rule by default `jax.numpy.trapz`). The condition number is estimated with `jax.numpy.linalg.cond`.

**Normalized Hamming distance.** We use the normalized Hamming distance computed on binary input matrices to compare the true system matrix $A$ with the empirically estimated matrix $\hat{A}$. To this end we first binarize $A$ and $\hat{A}$ via $B = \mathbb{I}_\tau(A)$ and $\hat{B} = \mathbb{I}_\tau(\hat{A})$ with threshold $\tau = 10^{-5}$ and where $\mathbb{I}_\tau$ is the indicator function that acts elementwise on matrix entries as

$$\mathbb{I}_\tau(a_{ij}) = \begin{cases} 1 & \text{if } a_{ij} > \tau; \\ 0 & \text{else} \end{cases}$$

The normalized Hamming distance between two matrices $B, \hat{B} \in \mathbb{R}^{n \times n}$ is then defined as $d_{\text{HMD}}(B, \hat{B}) = \frac{1}{n^2} \sum_{i,j} \mathbb{I}_{0.5}(B_{ij} \neq \hat{B}_{ij})$ where $B_{ij} \neq \hat{B}_{ij}$ has to be understood as a boolean comparison which evaluates to one under inequality and zero otherwise.

## D.3 EMPIRICAL ESTIMATORS

**SINDy.** SINDy, short for Sparse Identification of Nonlinear Dynamics (Brunton et al., 2016), is a widely adopted algorithm for system identification. It leverages a user-defined set of basis functions to execute $L_2$-regularized linear regression, mapping observations of the solution trajectory $\mathbf{x}(t_i)$ to their corresponding temporal derivatives $\dot{\mathbf{x}}(t_i)$. In practical applications, temporal derivatives are often unobservable, and SINDy estimates them through numerical finite difference approximations. We adopt the implementation available in `PySINDy` (de Silva et al., 2020), restrict the basis set to linear functions, and use the default optimization algorithm (sequentially thresholded least squares) which sets any coefficient whose magnitude falls below the user-defined threshold $\lambda$ to zero. Model and optimizer come with several hyper-parameters out of which we tune the $L_2$-regularization strength ($\alpha$), coefficient threshold ($\lambda$), finite difference approximation order and maximum number of iterations separately for each sample.

**Regularization of SINDy.** For each trajectory we select the pruning threshold $\lambda \in \{10^{-6}, \ldots, 10^{-1}\}$ that enforces the sparsity gate: after every ridge-regression step any coefficient whose magnitude falls below $\lambda$ is hard-set to zero, so increasing $\lambda$ enforces progressively sparser candidate systems. Complementing this, the ridge weight $\alpha \in \{0.001, 0.05, 0.1\}$ continuously shrinks the surviving coefficients toward the origin; larger values thus promote numerical stability without directly changing the zero pattern. For every trajectory, we select the optimal parameters $(\lambda, \alpha) = \text{argmax} R^2$ where the $R^2$ score is measured between the observed trajectory and the trajectory obtained by numerically solving the system estimate $\hat{A}$ for the observed initial value. Finally, the identifiability analysis is based on systems with well-fitted trajectories only, which we define as trajectories for which the estimate achieves $R^2 > 0.99$ and MSE $< 10^{-4}$. This regularization sets any coefficient that falls below $\lambda$ threshold to zero-hence aggressively promoting sparsity, which might lead to problems in low dimensional settings, see Fig. 4. In cases of very high sparsity and low system dimensionality, most coefficients of the true system matrix will be zero. In this case thresholding coefficients to zero biases the model towards a smaller Hamming distance. As the dimensionality increases or the sparsity reduces, this effect vanishes as there are multiple non-zero coefficients.

**Neural ODEs.** Neural ODEs (NODEs) (Chen et al., 2018) use a parameterized function $f_\theta(x(t))$ to approximate the dynamics underlying the observed trajectories. Instead of using finite difference schemes to estimate temporal derivatives, NODEs numerically integrate $f_\theta$ to obtain a solution that can be directly compared to the observed trajectory. In practice $f_\theta$ is implemented as a neural network; since we focus on linear systems, we use a model with multiple linear layers and no activation functions. To promote sparsity, we incorporate an L1 regularization term into the loss function. The total loss consists of the mean squared error (MSE) of the trajectories, augmented by the regularization parameter $\lambda$ multiplied by the L1 norm of the network's weights. To optimize the model we use the ode solvers implemented in `torchode` (Lienen & Günnemann, 2022), specifically the `Dopri5` solver in combination with the `IntegralController` for adaptive step size selection, with relative and absolute tolerances set to 1e-3 and 1e-6, respectively. Neural network parameters $\theta$ are optimized with PyTorch's `RMSprop` optimizer with a learning rate of 1e-3 to minimize the mean absolute error between the observed and predicted solution trajectory as in Chen et al. (2018). Optimization proceeds for 10000 iterations or until the loss falls below $10^{-5}$.

**Sparsity-regularization of Neural ODEs.** For the LinearNODE experiments we first identify well-fitted trajectories which we define as trajectories for which the empirical estimate (after numerical

integration from the ground truth initial value) achieves $R^2 > 0.99$ or MSE $< 10^{-4}$. The proportion of well-fitted trajectories (among all trajectories) for different system dimensionalities $n$ and sparsity levels $p$ is displayed for different values of regularization weight $\lambda \in \{0, 10^{-1}, 10^{-2}\}$ in Fig. 5. Among the $\lambda$-values that yield well-fitted trajectories, we then select the one that most faithfully reproduces the sparsity pattern of the ground-truth system matrix $A$. Specifically, for every trajectory we count the zero entries in the matrix estimate $\hat{A}$ and in the true $A$, compute the absolute difference in these counts, and use this sparsity-mismatch score as our selection metric. The optimal $\lambda$ for a given $p$ is selected as the value that minimizes the average sparsity-mismatch across all well-fitted trajectories. The effect of regularization weight $\lambda$, system dimensionality $n$ and sparsity $p$ on the sparsity-mismatch between model estimate $\hat{A}$ and ground truth system matrix $A$ is illustrated in Fig. 6. Our arguably very permissive model selection strategy reflects the idea that we are only interested in well-fitted models (as measured on the trajectory-level) in order to draw conclusions about (empirical) system identifiability rather than about optimization, model architecture or numerical issues.

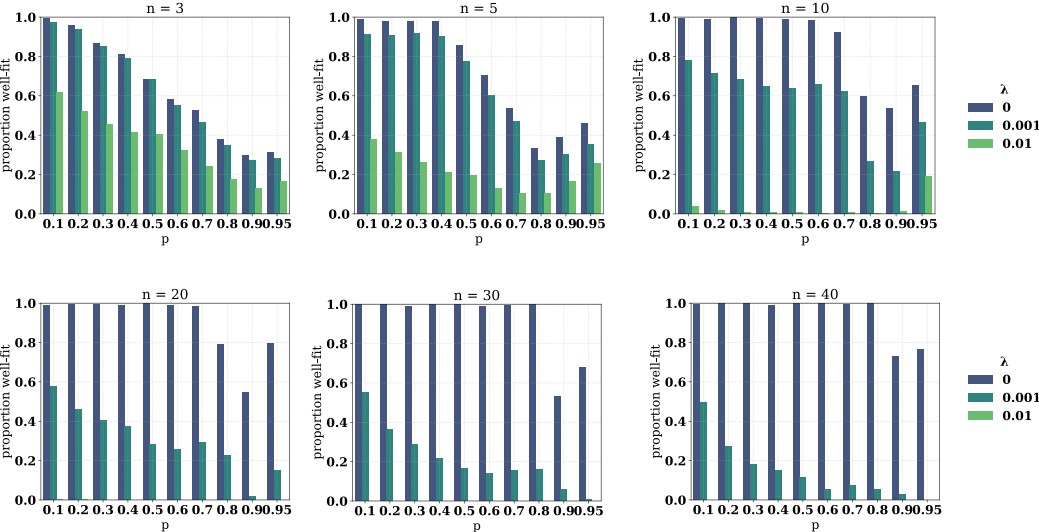

Figure 5: Proportion of trajectories that have been well-reconstructed by Sparse Neural ODEs for different regularization parameters $\lambda$ and different dimensions $n$ and sparsity levels $p$. For sparse systems, the model recovers only a smaller fraction of the trajectories.

### D.4 ADDITIONAL RESULTS ON TRAJECTORY-LEVEL IDENTIFIABILITY METRICS

We extend the results on trajectory-level identifiability metrics $d_A$ and SCN to a broad range of system dimensions $n$ and sparsity levels $p$. Box-plots for the two subgroups $A_{\sigma_2,\min}$ and $A_{\sigma_2,\max}$ introduced in Section 5.3 are displayed in Fig. 7 and Fig. 8. We observe the consistent trend that subgroup $A_{\sigma_2,\min}$, i.e., the subgroup with smaller second smallest singular value $\sigma_2$, leads to lower identifiability scores for both metrics (lower value for $d_A$ and higher value for SCN) in line with our theoretical results.

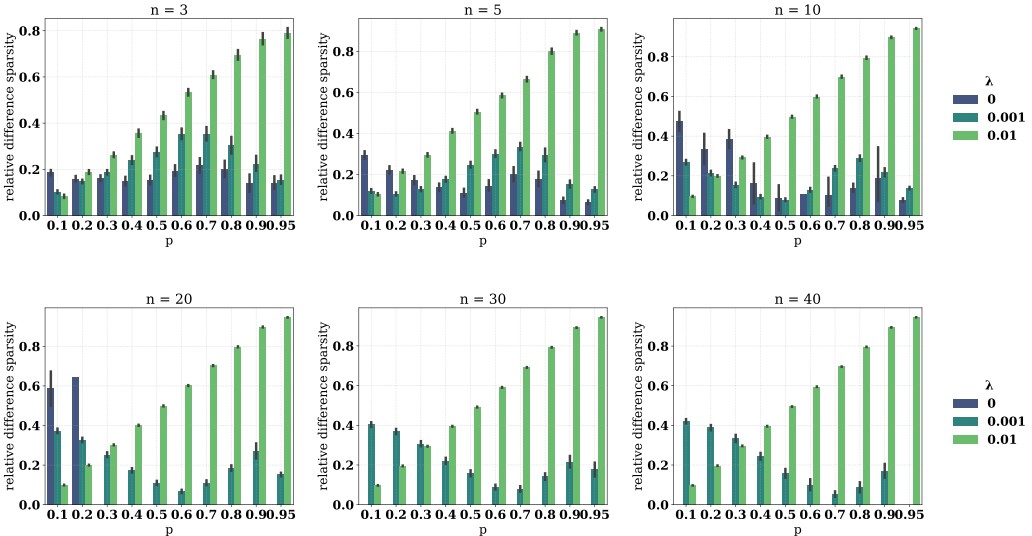

Figure 6: Relative difference in sparsity count (lower the better) between the true and reconstructed system matrices using Sparse Neural ODEs for different regularization parameters $\lambda$ and different dimensions $n$ and sparsity levels $p$. Lower regularization recovers dense matrices better, while higher values suit sparse ones.

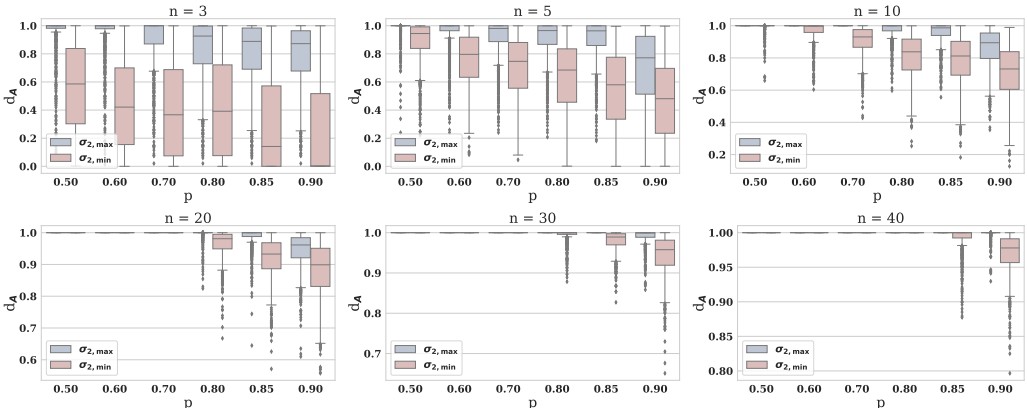

Figure 7: Box-plots of distance-to-unidentifiability $d_A$ for the least and most identifiable groups of systems for different p and $n$ values. Trajectories generated with $A_{\sigma_{2,\min}}$ lead to smaller $d_A$ than those produced with $A_{\sigma_{2,\max}}$.

**On the distance of a subspace of dimension $n$ from a random vector on the unit sphere.** We now empirically validate the close-to-unidentifiability metric $d_A$, which measures the distance between an initial condition $x_0$ and the kernel of the corresponding matrix $A$. Since we sample the initial conditions uniformly from the unit sphere, we can compare the empirical distribution of $d_A$ to the theoretical expected distance between a random unit vector and a $d_0$-dimensional subspace of $\mathbb{R}^n$. We partition the trajectories by the null-space dimension $d_0 = \dim(\ker(A))$ of their corresponding generating matrix $A$ and display the resulting distance-to-unidentifiability $d_A$ in Fig. 9. As expected, the (mean) empirical measure closely matches the theoretical expected distance between a random unit vector $\boldsymbol{x}_0$ and a $d_0$-dimensional subspace of $\mathbb{R}^n$ (Vershynin, 2018), given by

$$\mathbb{E}[d_A(x_0) \mid n, d_0] = \frac{\Gamma(n/2)\Gamma((n-d_0+1)/2)}{\Gamma((n-d_0)/2)\Gamma((n+1)/2)},$$

hence (in expectation) validating the computation of our distance-to-unidentifiability $d_A$.

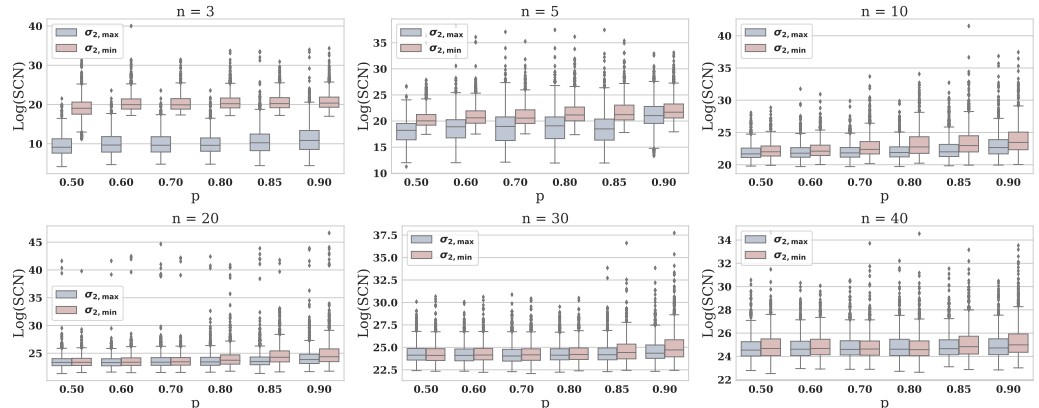

Figure 8: Box-plots of smoothed condition numbers (SCN) in log-scale for the least and most identifiable groups of systems for different p and $n$ values. Trajectories generated with $A_{\sigma_{2,\min}}$ lead to higher SCN than those produced with $A_{\sigma_{2,\max}}$.

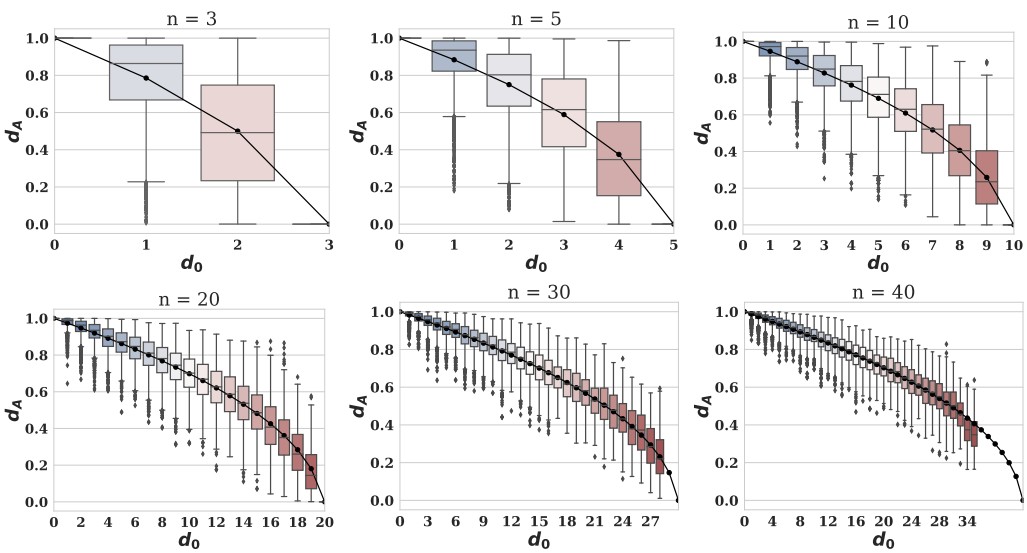

Figure 9: Box-plots of distance-to-unidentifiability $d_A$ different $n$ and different dimensions of $\ker(A)$ ($d_0 = \dim(\ker(A))$) together with the expected distance $\mathbb{E}[d_A(x_0) \mid n, d_0]$ (black line).

### D.5 ADDITIONAL EXPERIMENTS

A consequence of our findings in Lemma 4 is that the trajectories generated by two candidate matrices $A$ and $A'$ remain indistinguishable over longer time horizons when the initial condition lies closer to an invariant subspace. Here we examine how this behavior manifests in the performance of empirical estimators. To this end, we consider multiple configurations $(p, dim)$, where $p = \{0.1, 0.3, 0.7, 0.8, 0.9, 0.95\}$ and $dim = \{3, 5, 10, 20\}$. For each configuration we sample 10 matrices following the same data generation procedure as Section 5.3. We then generate four initial conditions at varying distances $d_A = \{0.1, 0.5, 1\}$ from the corresponding invariant subspace to assess how the estimators' performances change with increasing $d_A$. As described in Section D.1 we restrict the performance evaluation to well-fitted trajectories in order to investigate the effect of the distance on identifiability rather than model optimization properties. In Fig. 10, we display the Hamming distance and mean squared error (MSE) of the estimated matrices compared to the ground truth one. In line with our theory, the performance of both the Neural ODE as well as SINDy scales inversely with the distance to the invariant subspace, i.e., the smaller the distance $d_A$, the larger the estimation error.

A second factor that Lemma 4 predicts to affect empirical performance is the length of the observation interval: since trajectories remain indistinguishable up to time horizon T, reducing the observation window should degrade estimator performance. To investigate this aspect empirically, we use the same data from the previous experiment but restrict input to the estimator to only a fraction of the previous observation window. Our results in Fig. 11 show that indeed, as predicted, performance both in terms of Hamming distance as well as MSE decreases in almost all cases as the observation interval reduces. The exception to this observation occurs at the smallest distance to the invariant subspace, $d_A = 0.1$, where the performance remains constant across different trajectory lengths. As described before, $d_A = 0.1$ also leads to the largest errors overall, hence in this case the initial value is presumably already so close to the invariant subspace that unidentifiability even occurs at the longest observation window.

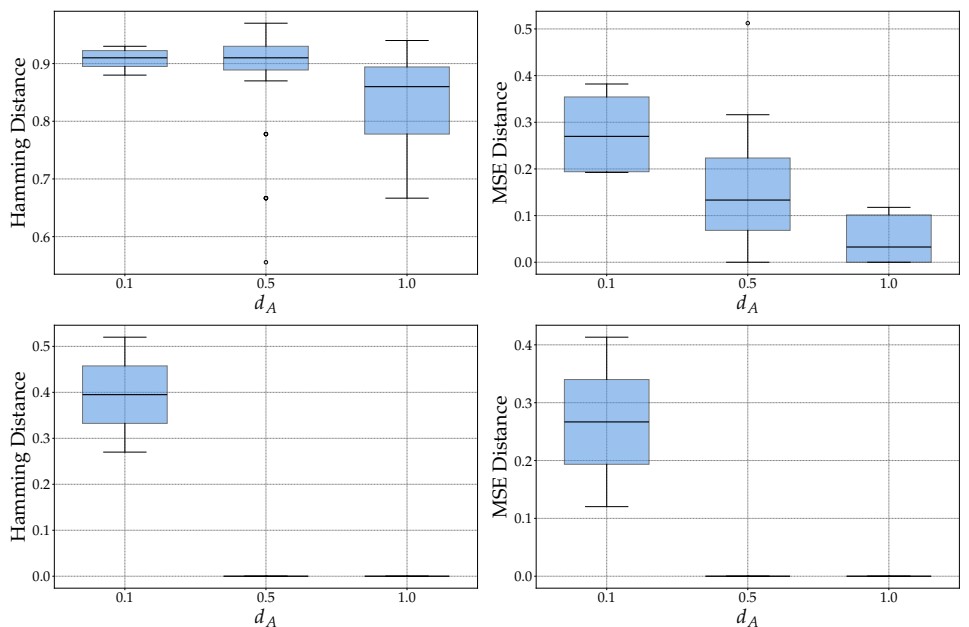

Figure 10: MSE and Hamming distance between the ground truth matrix and the matrix estimated from Neural ODE (top) and SINDy (bottom). The estimators' performance degrades with lower values of $d_A$.

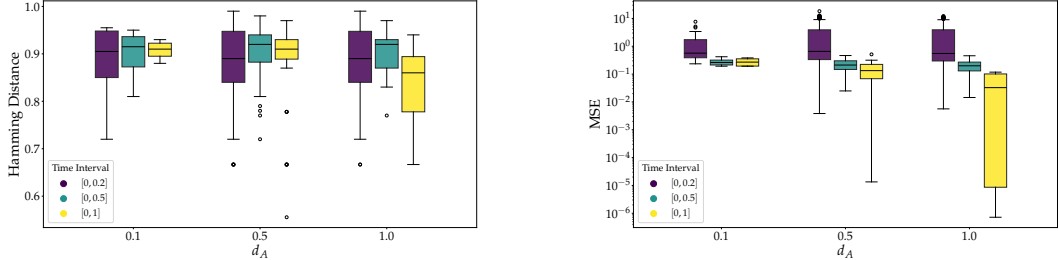

Figure 11: Hamming (left) and MSE (right) distance between the ground truth matrix and the matrix estimated from Neural ODE on trajectories spanning different time intervals.

### D.6 EXTENSION TO FURTHER MATRIX MODELS

We focus on further random matrix models and present here the results.

**Fixed number of zeros per row ensemble.** In this random matrix model we fix the number of zeros per row such that each row contains exactly $d(n) = \lfloor np \rfloor$ zeros. The remaining non-zero coefficients are sampled i.i.d. as $a_{ij} \overset{iid}{\sim} N(0, 1)$. We generate 100 system matrices and solve each of the for 100 initial values which are sampled uniformly at random from the unit circle in $\mathbb{R}^n$. Subsequently, we carry out the system-level identifiability and empirical identifiability analyses, following the same procedures described in the main text.

Results for both analyses are in line with the results reported in the main paper: system-level unidentifiability shows are sharp increase as sparsity increases (Fig. 12) and empirical identifiability shows a clear left-right gradient in Hamming distance for both SINDy (Fig. 15) and NODE (Fig. 16), confirming the expected rise in unidentifiability as sparsity increases.

**Sparse-Continuous ensemble with no zero rows or columns.** In this random matrix model we explicitly exclude matrices with zero rows or zero columns. For this, matrix entries are generated as $a_{ij} = g_{ij} b_{ij}$, where $b_{ij} \overset{iid}{\sim} Ber(p)$ and $g_{ij} \overset{iid}{\sim} N(0, 1)$. We again attempt to generate 100 system matrices, however, not all dimensionality $n$ and sparsity $p$ combinations permit any system matrices with no zero rows or columns. (E.g. a $3 \times 3$ matrix with sparsity level $0.9$ will have $\approx 8$ zeros and hence always multiple zero rows and/or columns.) We hence cap the number of attempts to generate a single matrix that fulfills the zero-rows / zero-columns constraints at 100 attempts. For every valid generated system matrix, we sample 100 initial values randomly from the unit circle in $\mathbb{R}^\ltimes$ and numerically solve the initial value problem to obtain 100 trajectories per system.

We perform system-level identifiability analysis as well as an empirical identifiability analysis. System-level metrics are provided in Fig. 13. Hamming-distances for different system dimensions $n$ and sparsity levels $p$ are reported in Fig. 15 for SINDy and in Fig. 16 for NODEs. For both estimators the average Hamming distance increases as sparsity rises, confirming the trends observed for other matrix models as well as the theoretical underpinnings.

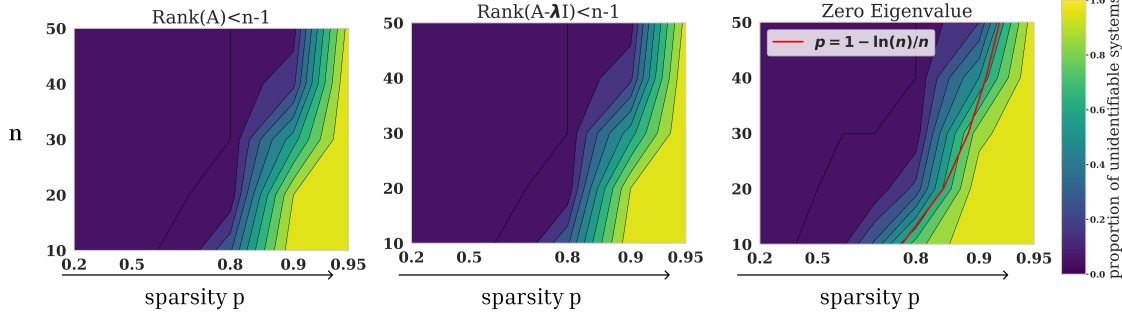

Figure 12: Proportion of matrices satisfying the conditions conditions $\mathrm{rank}(A - \lambda I) < n - 1$ (left), $\exists \lambda \in \mathbb{R} : \mathrm{rank}(A - \lambda I) < n - 1$ (center), and presence of zero eigenvalues (right) at different system dimensions $n$ and sparsity levels $p$ for **fixed number of zeros per row ensemble**.

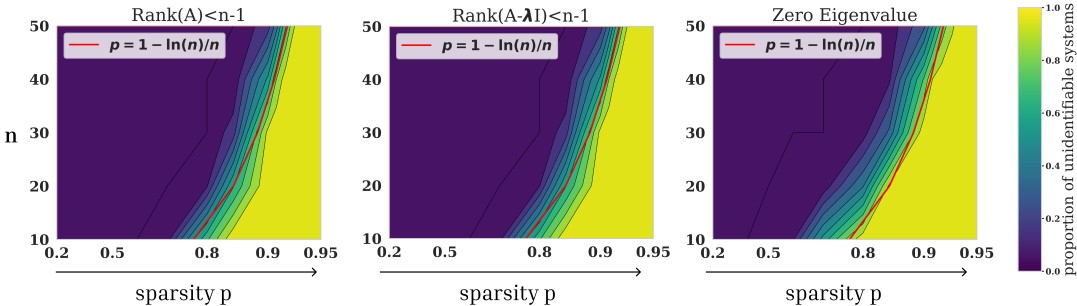

Figure 13: Proportion of matrices satisfying the conditions conditions $\mathrm{rank}(A - \lambda I) < n - 1$ (left), $\exists \lambda \in \mathbb{R} : \mathrm{rank}(A - \lambda I) < n - 1$ (center), and presence of zero eigenvalues (right) at different system dimensions $n$ and sparsity levels $p$ for **sparse-continuous ensemble with no zero rows**.

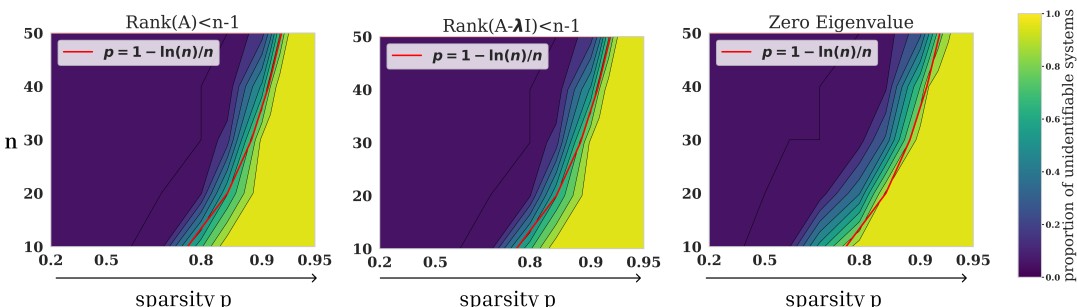

Figure 14: Proportion of matrices satisfying the conditions conditions $\mathrm{rank}(A - \lambda I) < n - 1$ (left), $\exists \lambda \in \mathbb{R} : \mathrm{rank}(A - \lambda I) < n - 1$ (center), and presence of zero eigenvalues (right) at different system dimensions $n$ and sparsity levels $p$ for **sparse-continuous ensemble with no zero columns**.

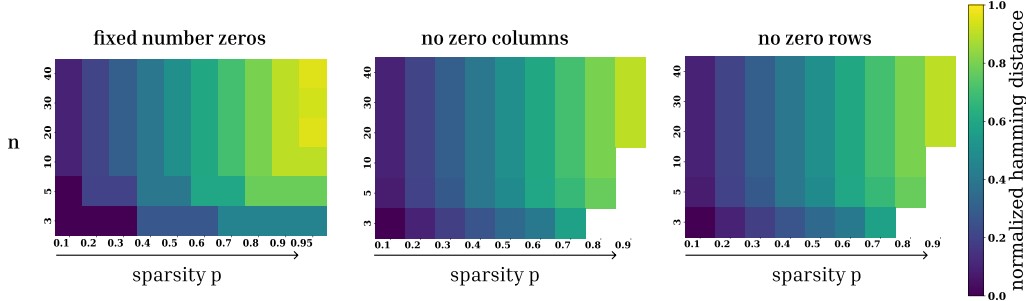

Figure 15: Hamming distance for different generating settings for SINDy on trajectories generated from **fixed number of zeros per row ensemble** (left), **sparse-continuous ensemble with no zero columns** (center), **sparse-continuous ensemble with no zero rows** (right).

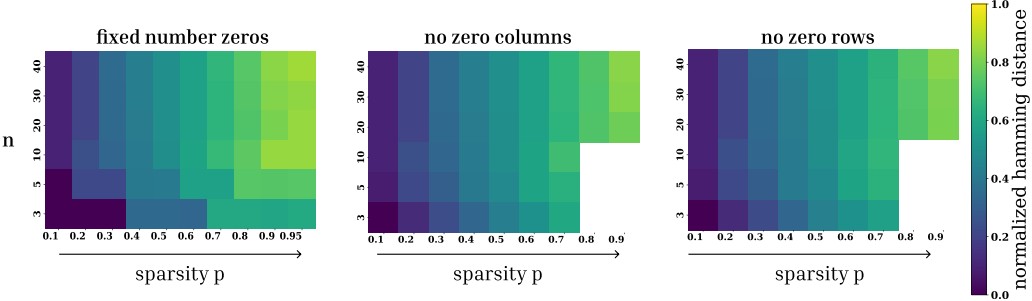

Figure 16: Hamming distance for different generating settings for NODE on trajectories generated from **fixed number of zeros per row ensemble** (left), **sparse-continuous ensemble with no zero columns** (center), **sparse-continuous ensemble with no zero rows** (right).

