# OpenReview forum: "Identifiability Challenges in Sparse Linear Ordinary Differential Equations"
_ICLR.cc/2026/Conference — ICLR 2026 Poster_

### Official Review · Reviewer_7pZ9 · 2025-10-16

**Soundness:** 3
**Presentation:** 3
**Contribution:** 3
**Rating:** 6
**Confidence:** 4

**Summary:**

This paper revisits the identifiability of linear ODE systems $dx(t)/dt = Ax(t)$ when the system matrix $A$ is sparse. The authors show that, unlike the dense case where almost all systems are identifiable from a single trajectory, sparsity introduces a positive probability of unidentifiability. They define a sparse–continuous ensemble and prove a sharp phase transition: when sparsity exceeds roughly $1-ln n /n$, systems become globally unidentifiable with high probability. The paper also introduces a trajectory-level metric to quantify how close a trajectory is to being unidentifiable. Simulations confirm that identifiability deteriorates with higher sparsity, both theoretically and in practice, using SINDy and Neural ODE estimators

**Strengths:**

1. Provides a clear theoretical characterization of when sparse linear ODEs lose identifiability.
2. Decomposing failure probability into global vs. trajectory unidentifiability (Eq. 2) is conceptually clarifying, and the distance $d_A(x_0)$ is a helpful practical proxy.
3. The sharp threshold result is elegant and connects to known results in random matrix theory.
4. Writing is well structured, and assumptions are transparently stated.

**Weaknesses:**

1. The strong assumptions (noise-free, single continuous trajectory, full observability) limit real-world applicability.
2. For readers less familiar with random matrix theory, it would be helpful to include a brief intuition box or paragraph below Lemma 3 explaining why the identifiability transition occurs precisely at $1-ln n/n$. A short, high-level explanation would make this elegant result more accessible and highlight its connection to classic random graph thresholds.
3. In the main text, the normalized Hamming distance is described as divided by $n$, but Appendix C.2 defines it as normalized by $n^2$. Please clarify which version was used, and make it consistent.

**Questions:**

1. In Fig. 4, for small dimensions (e.g., $n=3,5$) SINDy sometimes achieves low normalized Hamming distance even at very high sparsity. Could you comment on this regime?
2. In Section 2 (“Discussion of assumptions and limitations”), you might consider citing a related work on identifiability under hidden confounders: Wang et al. (2024) Identifiability analysis of linear ODE systems with hidden confounders.

---

> ### Author Response · Authors · 2025-11-21
> **Reply to review**
>
> We thank the reviewer for the comments and will address them one by one:
>
> - **Strong assumptions**: We refer the reviewer to our general response provided above. We appreciate the opportunity to elaborate on this point. Please let us know if this addresses your questions; we included it to the revised version in Appendix A.
>
> - **Threshold intuition**: From our perspective, there is not a direct intuition in our setting. The ln(n)/n threshold is commonly found in graph theory (closely related to sparse matrices when representing graphs via their adjacency matrix). There, one possibly intuitive way in which it comes up is the following: Consider a graph over $n$ vertices in which each pair of vertices is connected independently with probability $p$ via an edge. One can then ask when isolated vertices (not connected to any other vertex) remain in expectation. (This would correspond to zero rows/columns in our matrix setting.) Any given vertex is isolated if all the possible $n-1$ connections “failed”, i.e., with probability $(1-p)^{n-1}$. This can be approximated via $(1-p)^{n-1} = \exp(\ln((1-p)^{n-1})) = \exp((n-1) \ln(1-p)) \approx \exp(-p (n-1))$ for small $p$ (by using the Taylor expansion of $\ln(1-p) \approx -p + o(p^2))$). So for $n$ vertices, the expected number of isolated vertices is $n \exp(-pn)$. From this we see that $p=\ln(n)/n$ is the threshold that decides whether this expectation goes to infinity or 0 as $n$ grows. While this does not directly provide intuition for the threshold in our setting, it at least demonstrates on a mathematical level one way in which the $\ln(n)/n$ terms can come up from the approximation of $n (1-p)^{n-1}$.
> - **Hamming distance**: Thank you for catching this! It should indeed be $n^2$ in the main text as well. We corrected it in the revised version.
> - **SINDy in low dimensional settings**:  Thank you for this question and this detailed observation which has incentivized us to further look into these empirical results. While we believe that the results obtained by the NODE model (Figure 4, right) are consistent with the theoretical results, we agree that this is not entirely the case for SINDy (Figure 4, left). This behavior can be tied to the (default) sequential thresholded least-squared optimizer that is used to fit SINDy to the data, which we briefly comment on in the submission (Regularization of SINDy. l.901-904 of the old version, l.979-982 in the revised version). More specifically: Following least-squared optimization, this optimizer sets any coefficient that falls below a user-defined magnitude threshold to zero - hence promoting sparsity aggressively in the fitted model. In cases of very high sparsity and low system dimensionality, most coefficients of the true system matrix will be zero  (e.g. for the bottom row in the heatmaps of figure 4 with $n=3$ and $p = 0.95$, $3\times 3 \times 0.95 =~ 8.5$, i.e. there is often just a single non-zero coefficient). In this case thresholding coefficients towards zero biases the model towards a smaller Hamming distance. As the dimensionality increases or the sparsity reduces towards moderate sparsity levels, this effect vanishes as there are multiple non-zero coefficients.
>
> - **Additional reference**: Thanks a lot, we have incorporated this work along with other very recent works (e.g., https://arxiv.org/pdf/2511.08860) in the revised version.

---

> ### Comment · Reviewer_7pZ9 · 2025-11-24
>
> I thank the authors for their rebuttal. They have addressed all of my concerns, since my score is already positive, I will keep it.

---

> > ### Author Response · Authors · 2025-11-27
> > **Updated score**
> >
> > We are glad to see that we could answer all of your questions. We were also happy to see that you increased your score and would like to kindly ask what prompted you to lower it again and if there are any remaining questions.

---

### Official Review · Reviewer_h45Z · 2025-10-28

**Soundness:** 3
**Presentation:** 3
**Contribution:** 3
**Rating:** 6
**Confidence:** 3

**Summary:**

The paper presents a theoretical study of the identifiability of autonomous, linear and noise-free ODEs from a single trajectory, focusing on systems where the drift matrix is *sparse*. The authors distinguish between two notions of unidentifiability: (i) system-level (or global) unidentifiability, where the system is unidentifiable regardless of the initial condition, and (ii) trajectory-level unidentifiability, which occurs only for initial states lying in invariant subspaces of the system.

*For system-level unidentifiability*, they show that in sparse systems it is implied by rank deficiency (i.e., zero-eigenvalue degeneracy, Lemma 1), derive a lower bound on its probability via the occurrence of system matrices with two zero columns (Lemma 2), and extend the analysis (Lemma 3) using random graph theory to identify a dimensionality-dependent sparsity threshold governing unidentifiability.

*For trajectory-level unidentifiability*, they prove that the probability of “unlucky” initial conditions lying in invariant subspaces is zero, so identifiability depends solely on the system-level property. They further analyze near-invariant initial conditions, introducing a notion of distance to invariant subspaces, and showing that the indistinguishability time between trajectories of two systems agreeing on such a subspace increases inversely with this distance (Lemma 4).

Theoretical findings are finally supported by numerical experiments, which rely on neural network (Neural ODE) and symbolic regression (SINDy) methods.

**Strengths:**

1. The paper provides strong context through a clear discussion of related work, assumptions, and limitations. This situates the contribution well within the system identification and machine learning communities and makes its relevance easy to grasp.
2. The work extends classical results on the unidentifiability of linear systems to the practically important case of sparse systems, filling a notable gap in the existing theory.
3. The proofs are well structured and technically interesting.
4. The experimental section is well designed and directly supports the theoretical claims. The authors also apply two widely used system-identification methods (Neural ODE and SINDy) to test the practical implications of their results. The exposition of these empirical findings is clear and coherent (albeit limited, see Weekness 2 below).
5. The appendix is comprehensive, providing detailed proofs, additional empirical results, and clear information for reproducibility.

Overall, the paper is well written, technically sound, and contributes meaningful theoretical insight into the identifiability of sparse linear systems.

**Weaknesses:**

1. The most immediate limitation lies in the restriction to linear and noise-free systems. While the authors are transparent about this assumption, it considerably narrows the scope of applicability to real-world or data-driven settings.

2. The results on empirical unidentifiability (Section 5.3) are interesting but somewhat limited. The authors do not explicitly connect their theoretical findings on trajectory-level unidentifiability with their empirical unidentifiability results. For example, it would be interesting to examine how the two system-identification methods considered behave for initial conditions close to invariant subspaces, especially since the trajectory length should influence distinguishability in that regime (as suggested by Lemma 4).

3. While the use of a Bernoulli mask to model sparsity is theoretically convenient, the authors neither discuss nor acknowledge (or at least I don't find such a discussion/acknowledgement) that real-world sparse networks often exhibit "structured sparsity" (e.g., hub nodes or community structure). The Bernoulli mask (i.e. Erdős–Rényi) model does not capture such degree statistics, which may limit the generality of the conclusions.

**Questions:**

1. Why did you title Section 3 “Global Unidentifiability” instead of “System-level Unidentifiability”?
Put differently, why introduce the term system-level unidentifiability later in Section 3 rather than consistently using global unidentifiability throughout? This shift in terminology is somewhat confusing.

2. In Appendix B, you discuss the sparsity of real-world gene regulatory networks.
What are the node degree statistics of these networks, and can they be reproduced by your sparse–continuous random matrix model? If not, how might structured or heavy-tailed degree distributions affect your theoretical results?

3. Within your setup, is there a way to study how the closeness of an initial condition to an invariant subspace influences the performance of Neural ODE and SINDy? In particular, does the length of the observed trajectory or the inter-observation interval play a significant role in this regime, as suggested by Lemma 4?

4. In Figure 4, SINDy appears to match the true sparsity pattern almost perfectly for the 3-dimensional case across all sparsity levels.
How should these empirical findings be interpreted in light of your sharp sparsity threshold for global unidentifiability, and the corresponding empirical verification in Fig. 2? In other words, why does SINDy succeed in this low-dimensional setting even in regimes predicted to be unidentifiable?

---

> ### Author Response · Authors · 2025-11-21
> **Reply to review**
>
> We thank the reviewer for the review and for the useful comments. We address them one by one.
>
> - **Restriction to linear and noise-free systems**: since this point was raised by multiple reviewers, we refer to the general comment above. Please let us know if this addresses your questions; we included it to the revised version in Appendix A.
> - **“Global Unidentifiability”  vs “System-level Unidentifiability”**: Thanks for pointing out this inconsistency. We have consistently renamed it to “system-level unidentifiability” throughout the revised version of the submission.
> - **Sparsity distribution**: Indeed, the Bernoulli mask can be considered a canonical starting point that is also convenient to analyse.  We mention another sparsity model that breaks the independence of the zero structure across matrix entries in l.281 in the main text and provide more details in Appendix C.5 “Extensions to further matrix models”.  Concretely, we consider a model where zero rows and columns are not allowed and provide empirical results for these. We agree though that even more structured sparsity patterns are certainly an important and interesting direction for future work.
>
>   For the specific theoretical results, we highlight that Lemma 1 holds regardless of the concrete sparsity structure. The proof mentions the actual probability of a Bernoulli sparsity pattern $S$, but this concrete probability is not actually used/needed in the proof.
>
>   For Lemma 2 (non-zero positive probability of system level unidentifiability), we can make a more general statement as well: If the sparsity pattern puts non-zero probability on either at least two columns of all zeros, or at least three columns with just one non-zero entry in the same row, or at least four columns with just two non-zeros in the same rows, and so on, then there is still a non-zero probability that A is system level unidentifiable. In practice, this means that heavy tails of the degree distribution do not influence the core result (non-zero probability of system level unidentifiability) as long as there is still non-zero probability of multiple sparse columns.
>
>   Concretely, the technical result is the following: Let $P$ be a probability measure on $\{0,1\}^{n \times n}$ (i.e., any random sparsity pattern) such that for some $k\in \{0,1,\ldots,n-2\}$ the probability of the event E=“there are at least $k+2$ columns with at most $k$ non-zero entries in the same rows” under $P$ is non-zero, $P(E) = q > 0$. Then $A$ (following a sparse continuous ensemble with this new sparsity distribution) is not system level identifiable with probability at least $q>0$. We are happy to include this version in the paper if it is thought to be helpful.
>
> - **Additional results on empirical unidentifiability vs trajectory level unidentifiability for varying time intervals**: Thank you for suggesting these additional experiments. We performed additional experiments comparing the performance of the estimators for different distances to unidentifiability and different time intervals. We added the additional results in the Appendix section D.5.
>
> - **SINDy in low dimensional settings**: Thank you for this question and this detailed observation which has incentivized us to further look into these empirical results. While we believe that the results obtained by the NODE model (Figure 4, right) are consistent with the theoretical results, we agree that this is not entirely the case for SINDy (Figure 4, left). This behavior can be tied to the (default) sequential thresholded least-squared optimizer that is used to fit SINDy to the data, which we briefly comment on in the submission (Regularization of SINDy. l.901-904). More specifically: Following least-squared optimization, this optimizer sets any coefficient that falls below a user-defined magnitude threshold to zero - hence promoting sparsity aggressively in the fitted model. In cases of very high sparsity and low system dimensionality, most coefficients of the true system matrix will be zero (e.g. for the bottom row in the heatmaps of figure 4 with $n=3$ and $p = 0.95$, $3\times 3\times 0.95 =~ 8.5$, i.e. there is often just a single non-zero coefficient). In this case thresholding coefficients towards zero biases the model towards a smaller Hamming distance. As the dimensionality increases or the sparsity reduces towards moderate sparsity levels, this effect vanishes as there are multiple non-zero coefficients.

---

> > ### Comment · Reviewer_h45Z · 2025-11-24
> >
> > Thank you to the authors for the strong and detailed rebuttal.
> >
> > @*We are happy to include this version in the paper if it is thought to be helpful*: Yes, please do include it (either in the main text or in the Appendix).
> >
> > @*SINDy in low dimensional settings*: Can you also include this argument into the paper? Otherwise you leave the results in Fig 4 partially unaddressed.
> >
> > Other than these points, I am satisfied with the rebuttal. I will increase my score after the discussion period.

---

> > > ### Author Response · Authors · 2025-11-27
> > > **Updated revision**
> > >
> > > Thank you for your quick response, we are happy we could answer your questions. We have uploaded a second revised version, where we have included the explanation for SINDy’s performance in low dimensions and the clarification about alternative sparsity structures with a proof of the claimed result in the appendix.
> > >
> > > Since we understand that we will not be able to participate in any discussion after the official discussion period, we would like to kindly ask about the reason for delaying the updating of the score. Please let us know if any questions remain.

---

### Official Review · Reviewer_i8tC · 2025-10-31

**Soundness:** 3
**Presentation:** 3
**Contribution:** 3
**Rating:** 6
**Confidence:** 3

**Summary:**

The authors prove an identifiability result for a class of linear dynamics governed by a sparse-continuous random matrix. I went through the proofs in detail and they are reasonable, although this is outside my personal area of research so I can just say that the techniques and arguments seem reasonable. My only feedback is whether the authors can better motivate this class of sparse-continuous matrices and in what way they are relevant to machine learning. Of course identifying dynamics from time series is of interest, but a bit more discussion about why one should care about this is necessary.

**Strengths:**

Seems like a correct theoretical paper.

**Weaknesses:**

See  my response to the summary - the relevance of this class of matrices needs to be better motivated. I couldn't tell whether this is an impactful result and the authors chose this problem because its important, or if the authors just chose a class of matrices for which they knew how to prove something. if they can help me understand that better i will raise the score.

**Questions:**

no further questions

---

> ### Author Response · Authors · 2025-11-21
> **Reply to review**
>
> We thank the reviewer for their response and for the clarifying question. We appreciate the opportunity to elaborate further on this point refer the reviewer to our general reply above. We have also added it to the revised version in Appendix A. Please let us know if this addresses your concerns.

---

> > ### Author Response · Authors · 2025-11-27
> > **Further questions after the rebuttal?**
> >
> > Thank you again for your comments. We would like to kindly ask if our rebuttal (in particular the general response to all reviewers) answered your questions around the assumptions? Please let us know if any other questions remain.

---

### Author Response · Authors · 2025-11-21
**Practical relevance of our results**

All reviewers raised questions about the relevance of our problem setting and the assumptions we make. In particular, our assumptions of continuous, noise-free, fully observed trajectories have been regarded as limiting the scope/applicability of our findings. Indeed, those would be limiting if this was a method proposal, where the method only works on such observations. However, for analyzing unidentifiability, they are **the exact opposite of limiting**: Without these assumptions, no system would ever be identifiable, i.e., unidentifiability would be trivially given. Only once we have eliminated unidentifiability due to, e.g., undersampling, finite samples, and partial observations, can we make non-trivial claims about the inherent unidentifiability of the underlying systems.

Let us elaborate by splitting the discussion of our assumptions into two parts:
**(A)** The assumptions of **continuous** trajectories, **noise-free** observations and **full observability** are assumptions about our ability to observe/measure the system. For these, our setting corresponds to the “best case” idealized setting where (in principle) infinite data (both in terms of sampling frequency as well as in terms of noise instantiations) are available about all relevant parts of the system. As we’ll discuss one-by-one below, violating any of those assumptions will add additional identifiability issues, which are due to limitations of our observation process (not inherent to the system) that one could in principle overcome. Hence, instead of “our results only apply under those assumptions,” our results show unidentifiability to “still occur under these assumptions,” even when all solvable sources of unidentifiability due to limited measurement processes have been overcome. In short, these form the appropriate starting point for an identifiability analysis.
**(B)** The assumption of linear systems and sparsity are indeed assumptions about the underlying dynamics. We elaborate on the relevance of these assumptions below.

To summarize, by showing unidentifiability under those assumptions, our results transfer to all subsequent modifications for real-world modeling (but not vice versa) like discrete (scarce) observations, noise, non-linearities, partial observation. Therefore, our assumptions are not made for convenience (or in order to be able to prove something) but form the appropriate starting point for identifiability analysis.

 Let us go through the assumptions one-by-one for details.

**(A1) Continuous trajectories**: Any real-world measurement device will have a maximum sampling frequency, so all real-world data is necessarily discrete in time. Discrete observations add trivial unidentifiability issues around aliasing (cf. aliasing in the Nyquist sampling theorem). Hence, the continuous observation case is the best-case scenario - the limit of infinite sampling frequency - where we expect any remaining unidentifiability to be due to the fundamental problem setting, not our insufficient sampling rate.

**(A2) Noise-free observations**: Again, real-world data are almost always noisy. And again, in the presence of noise, system identification becomes inherently more difficult, namely probabilistic: multiple different systems may be compatible with the observed data, each with a different likelihood under the assumed noise model (for example, the standard assumption of i.i.d. additive Gaussian noise in classical machine learning often used to model “measurement errors”). Within this probabilistic framework, our noise-free setting can be interpreted as the best case scenario, where unidentifiability is not just due to suboptimal measurement devices. It can also be interpreted as the asymptotic limit of observing infinitely many trajectories for a single initial condition such that the “noise can be averaged out” to recover the noise-free setting. From this perspective, our results state under which conditions collecting more observations allows (in the asymptotic limit) the identification of the system. Or put differently, we show that there are cases in sparse systems where, even if more data was collected or where the noise level is reduced by other means, the true system remains unidentifiable.

---

> ### Author Response · Authors · 2025-11-21
> **Practical relevance of our results (part 2)**
>
> **(A3) Full observations**: Akin to discrete or noisy observations, only having partial observations of the system state will naturally render identifiability more difficult. Unlike discrete observations and noise, partial observations add a fundamental degree of unidentifiability as there is no “asymptotic limit” of more frequent or precise observations to recover our setting. Intuitively, when we allow for the presence of (an arbitrary number of) unobserved state variables, both the full system as well as the “observed part” (viewed as a submatrix of the full system) are inherently unidentifiable. As an extreme (and somewhat pathological) counterexample, one could imagine an exact copy of each observed variable among the latents such that all observed variables may either depend on the unobserved copy or the observed variable, which is indistinguishable from data. Hence, to study the identifiability under partial observations, one would have to provide additional limiting assumptions on the allowed type and nature of confounding. Exploring the palette of possible assumptions is an interesting direction for future work.
>
> **(B1) Linear systems**:
> There are two main reasons for this assumption: (a) Linear differential equation models are a core pillar of scientific modeling and widely used in practice due to their analytical tractability, relative simplicity, and inherent interpretability. While the real underlying dynamics of complex systems are arguably rarely perfectly linear, linear models also turn out to be surprisingly useful and predictive (akin to linear regression still being a competitive and reliable tool in many settings).
> (b) Non-linear ODEs are known to be unidentifiable from single trajectories if one does not impose heavy constraints on the allowed non-linear function family (Scholl et al., 2023) (with the notable exception of chaotic systems from just days ago: https://arxiv.org/pdf/2511.08860). Hence, the previous proclaimed state in the literature was a clear separation: “non-linear systems are unidentifiable while linear systems are almost surely identifiable.” Our work adds important nuances to the part about linear systems.
> Overall, the linear setting is both a practically relevant starting point as well as the most pressing one when identifiability is concerned.
>
> **(B2) Sparsity**: If it were somehow known that real-world systems never evolve according to dynamics below the sparsity-identifiability-threshold, our findings would indeed be of purely theoretical interest. However, the true sparsity of real-world dynamics remains inherently unknown. While it is impossible to determine whether a given unknown system is sparse, the examples in our submission indicate that this may not be unlikely in practice. Our results are thus important in that modeling practitioners should consider them when interpreting their results, e.g., despite a perfectly-fit model, the inferred weights might not reflect the true causal relations between variables due to unidentifiability issues.
> As a pragmatic step to highlight that high degrees of sparsity may occur in practically relevant settings, we examined published sparse systems, which we discuss in Section 5.1 (“Discussion on realistic sparsity levels”). In both reported cases, the published systems are on the sparser side of the threshold required for identifiability, which further supports the practical relevance of our findings. Nonetheless, we emphasize that the relevance of our results in terms of how frequently they show up cannot be empirically proven, since the notion of sparsity in our analysis refers to that of the true but unknown dynamics, rather than the model dynamics (models cannot be proven “true”). However, a practical implication for future method development efforts aimed at inferring the “ground truth” networks is for instance that such methods may produce estimates that fit the observed data equally well as the ground truth, yet are different. Importantly, this discrepancy might not stem from the limitations of the developed method itself, but rather from the nature of the underlying dynamics being estimated.
>
> **We would like to thank all reviewers for their comments and their encouragement to improve the motivation of our model class restrictions and the implications of our assumptions. We kindly ask if this clarifies the practical relevance to them - or to let us know which aspect needs further explanation**. We have updated the manuscript accordingly, you can find it in Section A of the revised version.

---

### Meta-Review · Area_Chair_zYCi · 2026-01-08

**Summary:**

The paper studies when you can identify an autonomous, linear, noise-free ODE from one fully observed continuous trajectory. Using sparse random-matrix models, it finds that:
- Sparser systems can be unidentifiable with non-zero probability, and there’s a sharp sparsity-dependent transition between identifiable vs. not.
- For an identifiable system, truly “bad” initial conditions happen with probability zero.
- But near-invariant subspaces can make trajectories look similar for a long time, so in practice identification may be delayed even when it’s theoretically possible.

**Reviewer Concerns:**

Reviewers mostly agree the paper is strong and the proofs match the experiments (Neural ODE, SINDy). They mainly questioned:
- How practical the results are, since the paper assumes ideal observations (fully observed, continuous, noise-free).
- Whether Erdos–Renyi/Bernoulli sparsity is realistic compared to structured sparsity in real systems.
- A few interpretation/presentation issues (SINDy’s low Hamming distance in low dimensions, Hamming normalization, terminology, and intuition for the \log(n)/n threshold).

The rebuttal largely resolves these by:
- Explaining the assumptions are best-case, so any unidentifiability is inherent.
- Fixing terminology and the Hamming normalization.
- Giving intuition for the \log(n)/n threshold (connected to isolated nodes).
- Extending discussion beyond i.i.d. sparsity and stating broader conditions.
- Adding experiments (sampling interval, distance-to-unidentifiability) and explaining SINDy’s behavior as a sequential-thresholding bias.

Overall, most key concerns were addressed.

**Reviewer Scores:**

- Reviewer i8tC: likely 6 ->8.  Their main issue was motivation/practical relevance, and the rebuttal addressed it.
- Reviewer h45Z: very likely 6 ->8. They said they’d raise the score after two specific fixes.
- Reviewer 7pZ9: no change (stays 6). They said concerns were addressed but would keep the score.

---

### Decision · Program_Chairs · 2026-01-26

Accept (Poster)